

# Forest disturbances and their impact on ground surface temperatures in permafrost-underlain forest in central Mongolia

Robin B. Zweigel[1,2], Dashtseren Avirmed[3,4], Khurelbaatar Temuujin[3], Clare Webster[1,5], Hanna Lee[6], Sebastian Westermann[1,2]

[1]Department of Geosciences, University of Oslo, Oslo, 0371, Norway
[2]Centre for Biogeochemistry in the Anthropocene, University of Oslo, Oslo, 0371, Norway
[3]Institute of Geography and Geoecology. Mongolian academy of Sciences, Ulaanbaatar, 15170 , Mongolia
[4]UNESCO Chair of Environmental Sciences in Eastern Central Asia, Mongolian Academy of Sciences, Ulaanbaatar, 15170, Mongolia
[5]Department of Geography, University of Zurich, 8057 Zürich, Switzerland
[6]Department of Biology, Norwegian University of Science and Technology, Trondheim, 7419, Norway

*Correspondence to*: Robin B. Zweigel (robinbz@uio.no)

**Abstract.** In the forest-steppe ecotone in central Mongolia, forest and permafrost exist close to their climatic limits and are co-located on north-facing slopes. The deciduous forest ecosystems and permafrost on these slopes are linked through interactions in the local energy and water balance. Furthermore, in this region the presence of such permafrost-forest systems provides essential services that supports local livelihoods and ecosystem function. However, forest disturbances that reduce or remove the forest canopies and lead to changes in surface cover could impact ground surface temperatures (GSTs) and potentially lead to permafrost degradation. In this study, we investigate the relationship between different forest states and GSTs at a site in the forest-steppe ecotone. We measured GSTs and surveyed vegetation density and surface cover over two years in an area that features both intact, dead and logged forest and dense stands of young regrowth. Overall, we find GSTs in summer and winter to vary substantially among the forest states, while differences in GST in spring and fall are small. Compared to the intact forest, the annual GST range is increased in the dead and logged forest while it is dampened in stands of young regrowth. Contrary to existing literature, we do not observe a general warming of the ground surface at disturbed sites, but instead find mean annual GSTs at disturbed sites to be 0.5°C lower than at intact sites. We also find substantial floor vegetation in the dead and logged forest, which has implications for livestock grazing patterns and remote sensing of forest disturbances.

**Short summary**

Two years of data along a forest disturbance gradient in Mongolia show a larger annual ground surface temperature range in dead and logged forests than intact forest, while the range is dampened in stands of young regrowth. Compared to intact





forest, mean annual ground surface temperatures are 0.5°C colder in dead and logged forest and dense stands of young regrowth. This is linked to differences in vegetation and surface cover due to the disturbance and patterns in livestock activity.

## 1 Introduction

Forest ecosystems regulate the energy and water balance of the ground below through process such as canopy shading, transpiration and accumulation of surface organic layers (Bonan and Shugart, 1989; Chasmer et al., 2011). As forest cover generally lowers ground temperatures and limits seasonal thaw depth, it provides the necessary conditions for permafrost persistence in many regions (Chang et al., 2015; Fisher et al., 2016). Major changes in forest structure and function thus have implications for the ground hydrological- and thermal regime, and potentially permafrost presence (Stuenzi et al., 2021a,

2022). Such disruptions to the forest cover can be biotic, for example insect infestations or diseases, or abiotic like wildfires, logging or windthrow, and we referred to them collectively as "forest disturbances". Previous work from sites affected by wildfire report reduced canopy shading, removal of surface organic layers and lower the surface albedo, which enhances absorption of solar radiation and gives higher ground surface temperatures (GSTs) (Klinge et al., 2021; Kopp et al., 2014; Li et al., 2021; Yoshikawa et al., 2002). Furthermore, the reduction in evapotranspiration following logging is found to elevate

soil moisture levels, which in turn increases active layer thickness and GSTs (Fedorov et al., 2017; Iwahana et al., 2005). However, most studies on forest disturbances in permafrost regions focus on summer aspects (Iwahana et al., 2005; Klinge et al., 2021; Kopp et al., 2014; Yoshikawa et al., 2002), and the impact of forest disturbances on year-round ground thermal regime remains unclear.

The forest-steppe ecotone in central Mongolia is at the southern border of both the Eurasian permafrost and boreal forest regions, where deciduous forests and permafrost is primarily found on north-facing slopes and their presences are intimately linked (Dashtseren et al., 2014; Ishikawa et al., 2018; Temuujin et al., 2024). Previous research suggests that forests cover in this region lowers ground temperatures and limits active layer depths, which promote permafrost presence. The existence of a shallow permafrost table in turn elevates soil moisture levels, which are essential for the survival of forests (Klinge et al.,

2021; Lange et al., 2015). The forested and permafrost underlain north-facing slopes are also important for local livelihoods and ecosystem function, as they provide essential wood, rangeland and water resources (Iijima et al., 2012; Karthe et al., 2013; Lkhagvadorj et al., 2013b, a). While the intertwined nature of permafrost-forest systems is widely documented (Dashtseren et al., 2014; Ishikawa et al., 2018; Temuujin et al., 2024; Undrakhbayar et al., 2024), their vulnerability to disturbances has received only little attention (Klinge et al., 2021). To this day, there is only sparse data on how forest

disturbances in the forest-steppe ecotone affect vegetation and surface conditions and associated GSTs.





In this study, we investigate forest disturbances and their effect on vegetation structure and density, surface cover and GSTs in Mongolia. We measure GSTs for 2 years and survey vegetation and surface cover in a study area in central Mongolia where different forest disturbance states (intact, dead, logged and young regrowth) occur at sites with similar topographic characteristics. Based on these observations, we seek to answer the following questions:

1. What are the effects of forest disturbances on vegetation densities in the canopy and forest floor, surface organic layers and snow cover?

2. How are annual and seasonal GSTs impacted by changes in forest and surface cover associated with forest disturbance?

## 2 Study area and methods

### 2.1 Site description

We collected field data over a period of 25 months at sites in the Bayanzurkh mountains in central Mongolia. Our study area (ca. 47.83°N, 107.22°E) is located in the forest-steppe ecotone in the southern part of the Khentii mountain range, 20 km east of Mongolia's capital Ulaanbaatar. The climate of the Ulaanbaatar region is highly continental, with large annual amplitude in temperature and overall low precipitation (Dalai et al., 2019). Temuujin et al. (2024) report mean annual air temperatures between -5°C to -2°C and an increase of 0.75°C over the period 1986-2017 at a similar site in the southern Khentii mountain range (Terelj, 1531 m a.s.l).

Our study area extends over three roughly parallel, north-east facing valleys that feature forest at different states of disturbance (Fig. 1). The upper parts of the western valley are covered with mature and intact forest dominated by larch (*Larix sibirica*), the most common forest tree in Mongolia (Dulamsuren et al., 2010). In the other parts of the study area, most trees have been killed by some disturbance and the dead tree trunks remain standing (Fig. 2b). While the exact type and timing of the disturbance is unclear, there are no visible signs of windthrow and burning, so biotic causes such as insect infestations or diseases seem likely. In parts of the disturbed area there is regrowth of various young deciduous trees (larch, birch and aspen), especially in the north-facing part of the central valley (Fig. 2b), as well as some highly local, dense forest stands in topographic concavities (Fig. 1). In both the intact and disturbed forests the vegetation consists of a canopy layer that is made up of intact, dead and young trees, and floor vegetation consisting mostly of grasses and dwarf shrubs. The upper ridge of the central and eastern valleys is accessible by car from the south, and the forest has largely been logged circa 150 m downslope of the ridge (Fig. 2a & c). The area is inhabited by semi-nomadic herders who make their camps on south-facing slope, with the closest camp found 700m south of our study area.





**Figure 1: (a)** Overview of the main landcover and topographic characteristics of the study area in the Bayanzurkh mountains, as well as the location of the GST loggers and the air temperature measurement. The coloured markers for the loggers are grouped by the apparent forest states. Background imagery from 10. September 2022 (Esri, 2023) and contour lines derived from the SRTM 30 m Digital Elevation Model (NASA JPL, 2013). **(b)** Location of the study area (red triangle) within Mongolia (national boundaries from Runfola et al. (2020)).

Within the study area we select 10 field sites targeting the four distinct forest states found in the study area: intact forest, dead forest, logged forest and stands of young regrowth (Table 1 & A1). The field sites are chosen to have broadly similar topographic characteristics (Table 1) and to avoid transitional areas between the forest states. INTACT_1 and INTACT_2 are in the intact forest in the upper part of the western valley. While the forest at both INTACT_1 and INTACT_2 forms a canopy without larger gaps, we note scattered logging and a lack of young trees at these sites. DEAD_1, DEAD_2 and DEAD_3 are in the dead forest with standing tree trunks in the lower parts of the western and central valleys (Fig. 1), where



the forest floor in summer is covered by abundant grass vegetation (Fig. 2b). In the surroundings of DEAD_3, there is also some scattered, young regrowth of birch and larch. YOUNG_1 and YOUNG_2 are situated in dense stands of young aspen and larch, respectively. These stands are only 20-30 m in diameter, and thus represent highly local conditions that are in stark contrast to that of the neighbouring dead forest at DEAD_2 and DEAD_3. To emphasize their small spatial extent, the young regrowth is referred to as "stands" rather than "forest". LOGGED_1, LOGGED_2 and LOGGED_3 are in the logged forest in the upper part of the central and eastern valleys (Fig. 1a). While LOGGED_1 and LOGGED_3 only feature grass vegetation, there are also scattered shrubs at LOGGED_2 (Fig. 2c). In addition to these field sites we measure air temperatures in a relatively flat area between the western and central valley at 1810 m a.s.l. (Fig. 1a).



**Figure 2: (a) 25. June 2022: Looking north from the saddle point close to LOGGED_3, showing the transition from logged and grazed areas in the foreground, to scattered, young regrowth mixed with dead tree trunks in the lower parts of the central valley. The intact and dead forest in the western valley is visible above the ridge in the background. (b) 19. August 2023: Dead forest and dense floor vegetation in the vicinity of DEAD_1. (c) 19. August 2023: Logged forest at LOGGED_2 with floor vegetation consisting of grasses and shrubs (knife handle 10 cm for scale).**





**Table 1: Topographic characteristics for the field sites derived from the SRTM 30 m Digital Elevation Model (NASA JPL, 2013). Aspect is in degrees counterclockwise from south following the notation by Dozier and Frew (1990) and the cardinal/ordinal direction is indicated in brackets. SVF denotes the sky view fraction, i.e. the fraction of the upward facing hemisphere not obstructed by terrain.**

| Field site | Elevation [m a.s.l.] | Slope [°] | Aspect [°] | SVF [-] |
|---|---|---|---|---|
| INTACT_1 | 1778 | 14.7 | 141 (NE) | 0.95 |
| INTACT_2 | 1808 | 17 | 138 (NE) | 0.95 |
| DEAD_1 | 1690 | 18.7 | 117 (NE) | 0.94 |
| DEAD_2 | 1686 | 17.1 | 128 (NE) | 0.94 |
| DEAD_3 | 1690 | 18.9 | 172 (N) | 0.94 |
| YOUNG_1 | 1686 | 17.9 | 128 (NE) | 0.94 |
| YOUNG_2 | 1692 | 19.3 | 167 (N) | 0.94 |
| LOGGED_1 | 1748 | 14.8 | 131 (NE) | 0.95 |
| LOGGED_2 | 1742 | 15.8 | 131 (NE) | 0.95 |
| LOGGED_3 | 1775 | 8.1 | 187 (N) | 0.96 |


## 2.2 Temperature measurements

We measure temperatures using iButton (© Maxim) temperature loggers that we install at the field sites in summer 2022. We use iButtons of the type "DS1922L", which feature an operating range from -40 to +85°C, and a numerical resolution and accuracy of 0.0625°C and 0.5°C, respectively (Maxim Integrated, 2015). These loggers are small and are placed below any

layers of surface organics (litter and/or moss) to measure near-surface ground surface temperature (GST). Similar setups using iButton loggers to measure GSTs have successfully been conducted at grassland- (Zweigel et al., 2024a), rockwall- (Schmidt et al., 2021) and high-arctic sites (Zweigel et al., 2021). The loggers are set to record at 4-hourly intervals, from which we compute the daily temperatures used in this study. For all sites we successfully obtain a complete 25-month record of daily temperature spanning from 10. August 2022 to 19. September 2024, containing the hydrological years 2023 and

2024. Note that unless specified otherwise, the GSTs and temperature-based indices presented in this study are average values for all loggers within the different forest states.

From the daily temperature records we derive temporal mean GSTs and various temperature-based indices. The mean annual GST, MAGST, is calculated over the hydrological years 2023 and 2024, while seasonal GSTs are calculated over 3-month

periods for fall, winter, spring and summer (SON: September, October and November; DJF: December, January and February; MAM: March, April and May; JJA: June, July and August, respectively). In addition to average GSTs, we use the measured air temperatures and GSTs to calculate the surface offset ($SO$) and the scale factors for freezing ($n_f$) and thawing ($n_t$). The surface offset is given as the difference between the mean annual temperatures of the ground surface and the air (Smith and Riseborough, 2002):





$$SO = MAGST - MAAT$$

( 1 )

Where MAAT is the mean annual air temperature over the same hydrological year as MAGST. A positive SO indicates that on annual scales, processes related snow and vegetation cover give higher ground surface temperatures than air temperatures. The scale factors for freezing and thawing are calculated following (Smith and Riseborough, 1996):

$$n_f = \frac{FDD_{air}}{FDD_{surface}}$$

( 2 )

$$n_t = \frac{TDD_{air}}{TDD_{surface}}$$

( 3 )

Where *FDD* and *TDD* denote the freezing and thawing degree days defined as:

$$FDD = \sum_{i=1}^{n} |T_i|, \qquad T_i < 0°C$$

( 4 )

$$TDD = \sum_{i=1}^{n} |T_i|, \qquad T_i > 0°C$$

( 5 )

Here, $T_i$ is the daily averaged temperature of the air or ground surface, and *n* is the number of days per hydrological year. $n_f$
and $n_t$ typically vary between 0 and 1, with values close to 1 indicating that air and ground surface temperatures are closely related in the cold and warm season, respectively, while low values indicate that they are more disconnected. Note that throughout this study we refer to mean GSTs and temperature indices by the year in which they end, i.e. SO (2024) in Table 2 is the surface offset over the hydrological year 2024 from September 2023 through August 2024.

## 2.3 Vegetation density

We quantify vegetation density by estimating the Plant Area Index (PAI) at our field sites from hemispherical images, similar to Zweigel (2024a). The PAI is the total one-sided area of plant material per unit of ground surface and includes both the contribution from live foliage (leaves/needles) and that of woody elements such as stems and branches (Liu et al., 2021). To calculate the PAI from hemispherical images, pixels of vegetation and sky need to be distinguished, and the area of plant material per ground area that would yield such a distributing of vegetation and sky can be calculated. In this study, we use
the software Hemisfer (Schleppi et al., 2007) and the methodology by Lang (1987), as well as corrections for slope (Schleppi et al., 2007) and canopy clumping (Chen and Cihlar, 1995), to derive PAI at all sites. While other software/algorithms to derive PAI exist (Liu et al., 2021), and the exact value is influenced by illumination conditions and





assumptions regarding vegetation properties (e.g. Thimonier et al., 2010; Zhang et al., 2005), PAIs that are calculated using the same procedures provide a consistent and quantitative measure of vegetation density that can be compared among sites.


At forested sites, the PAI is usually calculated from hemispherical images of the forest canopy only, but in our study area there is also substantial floor vegetation especially at disturbed sites (Sect. 2.1). We thus calculate the PAI from hemispherical images taken at two levels: (1) 0.5 m above the ground surface and (2) at the ground surface. The former is referred to as "$PAI_{canopy}$" as it captures the forest canopy only, while the latter includes both the contributions of the forest

canopy and floor vegetation and is denoted "$PAI_{total}$". The hemispheric images from which the PAIs in this study are derived were acquired on 22. December 2022 , 6. March 2023 and 19. August 2023.

## 2.4 Surface cover

We surveyed the surface conditions in the field area in summer and winter. During field visits on 19. August 2023 and 17. September 2024 we measured the height of floor vegetation and the presence of surface organic layers at the ground surface.

In winter 2023 we surveyed the snow cover at all sites on 22. December 2022 and 6. March 2023. At the visit in early winter, we took measurements of snow height at each site. In late winter, we observed livestock trampling of the snow cover at the field sites, and we took several measurements of snow height at each site to capture the apparent small-scale variability. At the field visit in late winter, we also investigated the presence of litter layers at the base of the snowpack at all sites. Measurements in summer focussed on the immediate vicinity (< 20 cm) of the GST logger location, while observations in

winter were generally taken 0.5-2 m away to avoid disturbing the snow cover at the logger location.

## 3 Results

### 3.1 Ground surface temperatures

We observe different seasonal evolution of GSTs for the four forest states in the Bayanzurkh mountains. The variability of GSTs with forest state is largest in summer and winter, while GSTs are rather similar among the different forest states in

spring and fall (Fig. 3b & B1). During summer, the GSTs in the dead forest and stands of young regrowth are rather similar to those in the intact forest, while the GSTs in the logged forest are circa 1.5°C higher in both 2023 and 2024. In the cold season the differences in GSTs among the forest states are less systematic (Fig. B1), but overall the intact forest and stands of young regrowth feature winter GSTs 2 – 3°C higher than the dead and logged forest (Fig. 3b). The largest range in seasonal GSTs is found in the logged forest, spanning from -13.9°C to +13.4°C in 2023 (Fig. 3b). In both years the disturbed

forest states and the stands of young regrowth feature rather similar MAGSTs, which are 0.5°C higher than in the intact forest (Fig. 3a). Comparing the two years shows a lower annual GSTs range and higher MAGSTs in 2024 than in 2023 for all forest states, with the latter driven mostly by warmer GSTs in fall and winter (Fig. 3b). The variability in seasonal GSTs among the 2-3 temperature loggers at each site are generally small (<1°C) compared to the GST differences among forest





states (Fig. 3b). The largest differences of >2°C are observed in summer in the logged forest where we also find the largest

spread in PAI$_{total}$ (Sect. 3.2), and in winter and spring for sites subject to extensive snow trampling (Sect. 3.3, Table C2).

Similarly, the variability of MAGSTs within each forest states is generally small with the largest values of 1.5-2.2°C found

in the logged forest.

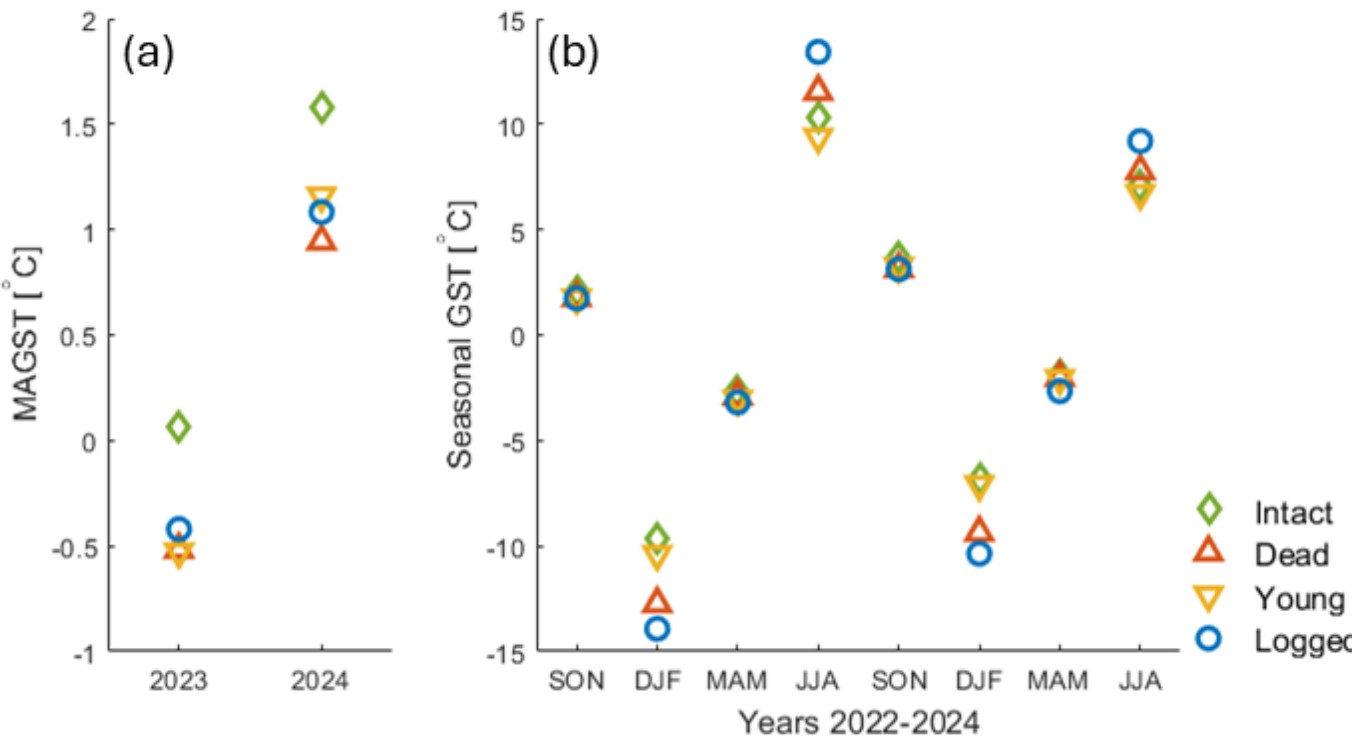

**Figure 3: (a) Mean annual and (b) seasonal GSTs for the different forest states. SON: September, October and November. DJF:**
**December, January and February. MAM: March, April and May. JJA: June, July and August. Note the different scales on the**
**temperature axes of (a) and (b). Daily mean GST and GST range for the different forest states are presented in Figs. B1 and B2.**

Comparing monthly GSTs from the logged and dead forest to the intact forest sites shows clear seasonal differences (Fig. 4).

We find GSTs in the logged and dead forest to be higher than in the intact forest in summer while they are lower in winter.

The strongest difference between disturbed sites and intact forest is found in the logged forest, where monthly GSTs are up

to -5.1°C lower and +3.7°C higher than those in the intact forest (Fig. 4). In the dense stands of young regrowth we measure

similar or lower GSTs than in the intact forest for most months, except for a short period in early winter in both years (Fig.

4).





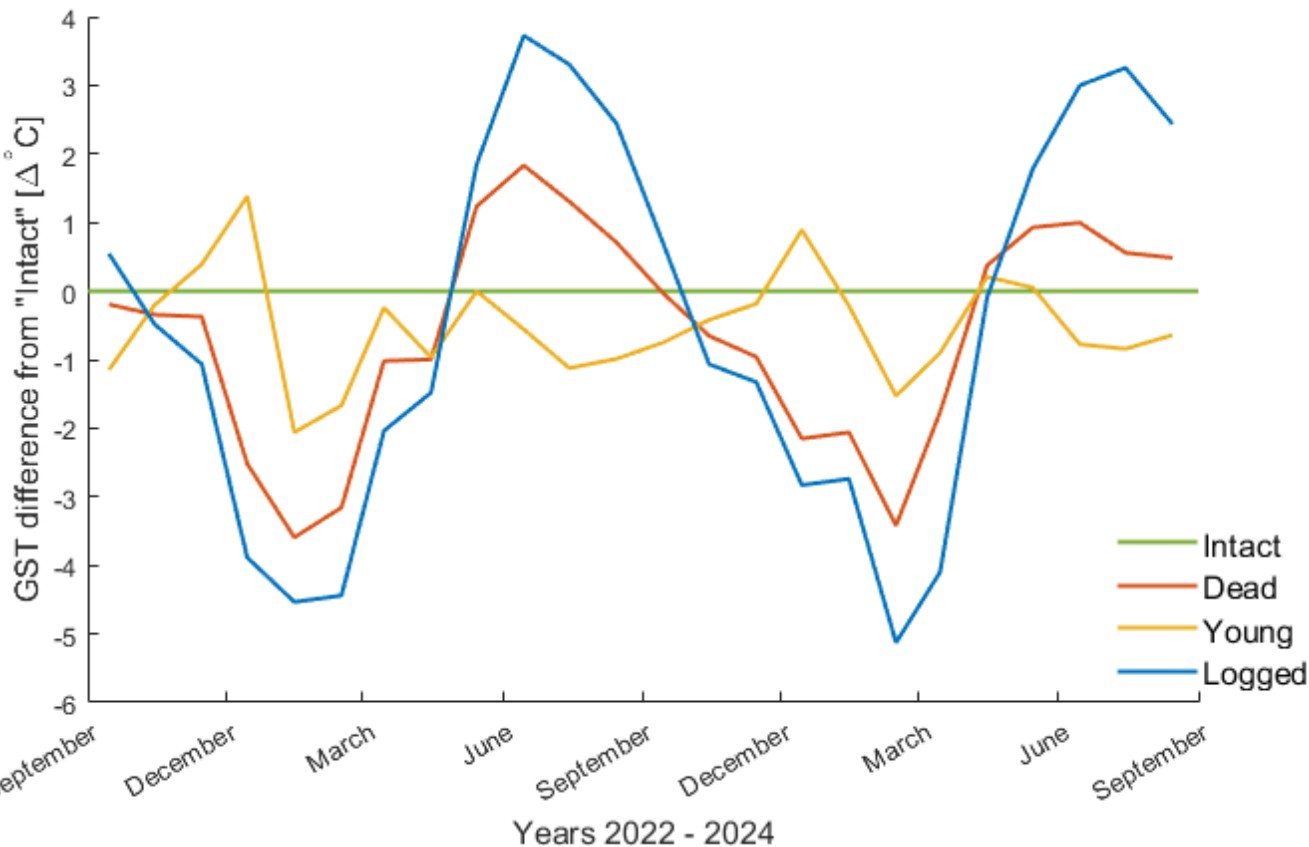

**Figure 4: Difference in monthly GST between the disturbed forest states and the intact forest. Positive values indicate that GSTs in**
**the disturbed forest are higher than in the intact forest.**

In Table 2 we present metrics relating the GST of the different forest states to measured air temperatures. The surface offset
(Eq. 1) is close to zero or positive for all forest states in both years, indicating that on annual timescales GSTs are generally
higher or equal to air temperatures. Furthermore, the different disturbed forest states have rather similar surface offsets that
are circa 0.5°C lower than for the intact forest in both years. The largest scale factors for both freezing (Eq. 2) and thawing

(Eq. 3) are found at the logged forest, which imply that GSTs at these sites more closely follow air temperatures. In both
years, the intact forest has the smallest scale factor for freezing while the stands of young regrowth have the smallest scale
factors for thawing, indicating that these forest states feature the GSTs that are most disconnected from air temperatures in
winter and summer, respectively. Overall, surface offsets are higher and the scale factor for freezing is lower for all forest
states in 2024, while the scale factor for thawing is rather similar among the years (Table 2). This indicates that winter rather

than summer effects contributed to the higher MAGSTs measured in 2024 (Fig. 3a).



**Table 2: Temperature metrics for the different forest states calculated over the hydrological years 2023 and 2024. SO: Surface offset (Eq. 1), $n_f$: scale factor for freezing (Eq. 2), $n_t$: scale factor for thawing (Eq. 3).**

| Forest state | SO (2023) | SO (2024) | $n_f$ (2023) | $n_f$ (2024) | $n_t$ (2023) | $n_t$ (2024) |
|---|---|---|---|---|---|---|
| Intact | 0.55 | 1.05 | 0.58 | 0.44 | 0.64 | 0.66 |
| Dead | -0.03 | 0.41 | 0.74 | 0.59 | 0.71 | 0.69 |
| Young | -0.05 | 0.63 | 0.62 | 0.47 | 0.58 | 0.61 |
| Logged | 0.06 | 0.55 | 0.83 | 0.68 | 0.82 | 0.80 |

### 3.2 Vegetation density

The $PAI_{canopy}$ values calculated for the forest states show distinct patterns in vegetation density. Figure 5a shows how the $PAI_{canopy}$ in the intact forest and stands of young regrowth, which both feature live canopies, increases substantially from winter to summer. Particularly, the $PAI_{canopy}$ for the stands of young regrowth increases by 6.3 times from late winter to summer, which is markedly more than the increase in $PAI_{canopy}$ of 4.6 times for the intact forest. We also observe $PAI_{canopy}$ in the intact forest to decrease during winter, which is not observed at any other forest states. This could be linked to gradual

shedding of needles, consistent with observation of needles at the snow surface in late winter, or to uncertainties in the PAI calculation under different illumination conditions (see Sect. 4.1). Nevertheless, the difference in wintertime $PAI_{canopy}$ in the intact forest is small (0.5) compared to the PAI increase from early and late winter to summer of 2.2 and 2.7, respectively.

We observe no seasonal evolution in $PAI_{canopy}$ in the dead and logged forest, in line with the absence of live canopies at these

sites. In the dead forest we calculate a $PAI_{canopy}$ of 0.4 in winter, which is lower than for intact forest but still a substantial increase compared to the absent canopy of the logged forest (Fig. 5a). In the logged forest the wintertime estimates reveal that the remaining tree stumps do not contribute to $PAI_{canopy}$ (values $\ll$ 0.1), and we thus set $PAI_{canopy}$ to 0 also in summer. In the dead forest $PAI_{canopy}$ increases to 0.6 in summer, which is due to scattered regrowth of young trees at the "DEAD_3" site (Sect. 2.1), while the other sites in the dead forest show no increase in $PAI_{canopy}$ in summer (Table C1).


Several of the forest states also feature substantial floor vegetation, which is evident when comparing summertime $PAI_{canopy}$ and $PAI_{total}$ (Fig. 5b). As all forest states feature some floor vegetation, $PAI_{total}$ is greater than $PAI_{canopy}$ at all sites, with an increase in PAI of 1.1 in the intact forest. The smallest increase is found in the stands of young regrowth (0.7), while the largest increase is in the dead forest (3.9). For the logged forest where there is no canopy, we calculate a $PAI_{total}$ of 1.6 which

stems entirely from floor vegetation. The variability of $PAI_{total}$ within each forest state is small compared to their absolute value, except for the logged forest where $PAI_{total}$ ranges from 0.9 to 2.5 (Table C1).





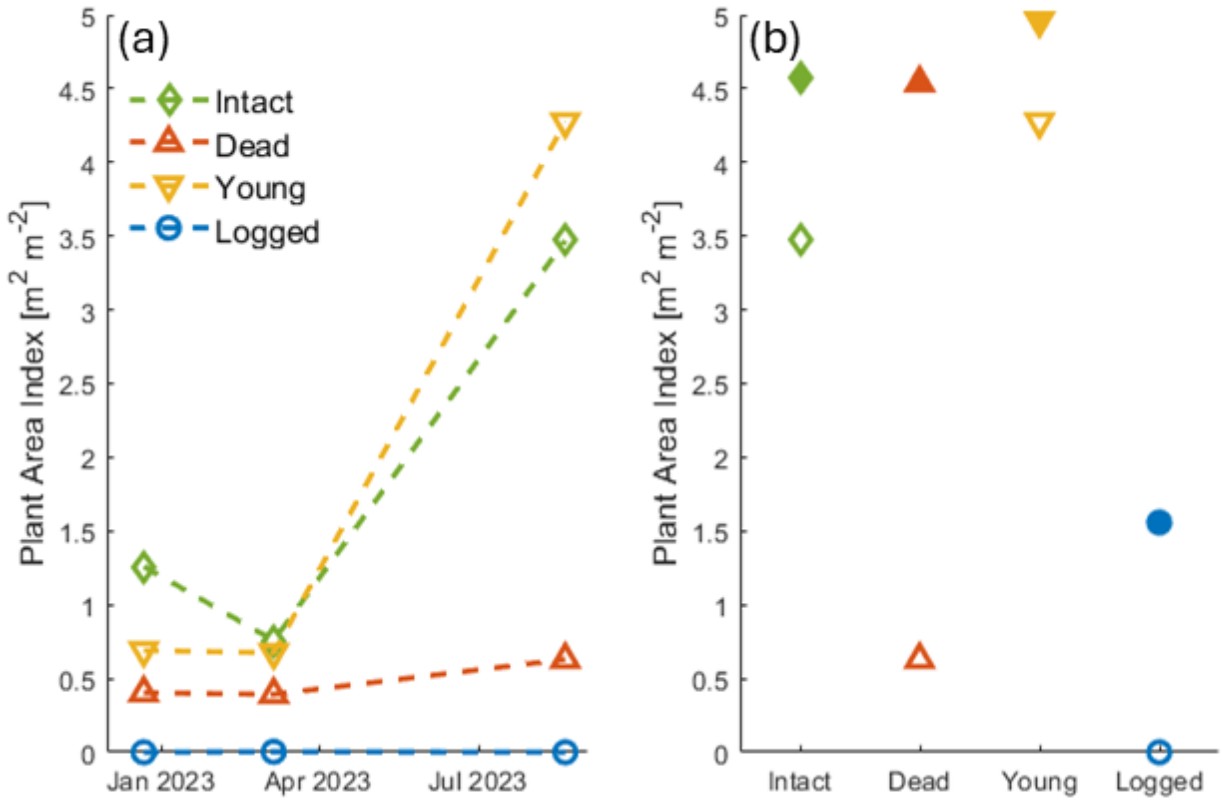

**Figure 5: (a) Evolution of PAI$_{canopy}$ for the different forest states in 2023. (b) Comparison of PAI$_{canopy}$ (open markers) and PAI$_{total}$ (filled markers) on the 19th August 2023**

Furthermore, we compare vegetation density as quantified by PAI$_{canopy}$ to the scale factors for freezing and thawing (Fig. 6). The scale factors for thawing and freezing are lower for higher summertime and wintertime PAI$_{canopy}$, respectively, which indicate that GSTs are more disconnected from air temperatures as sites with denser canopies.





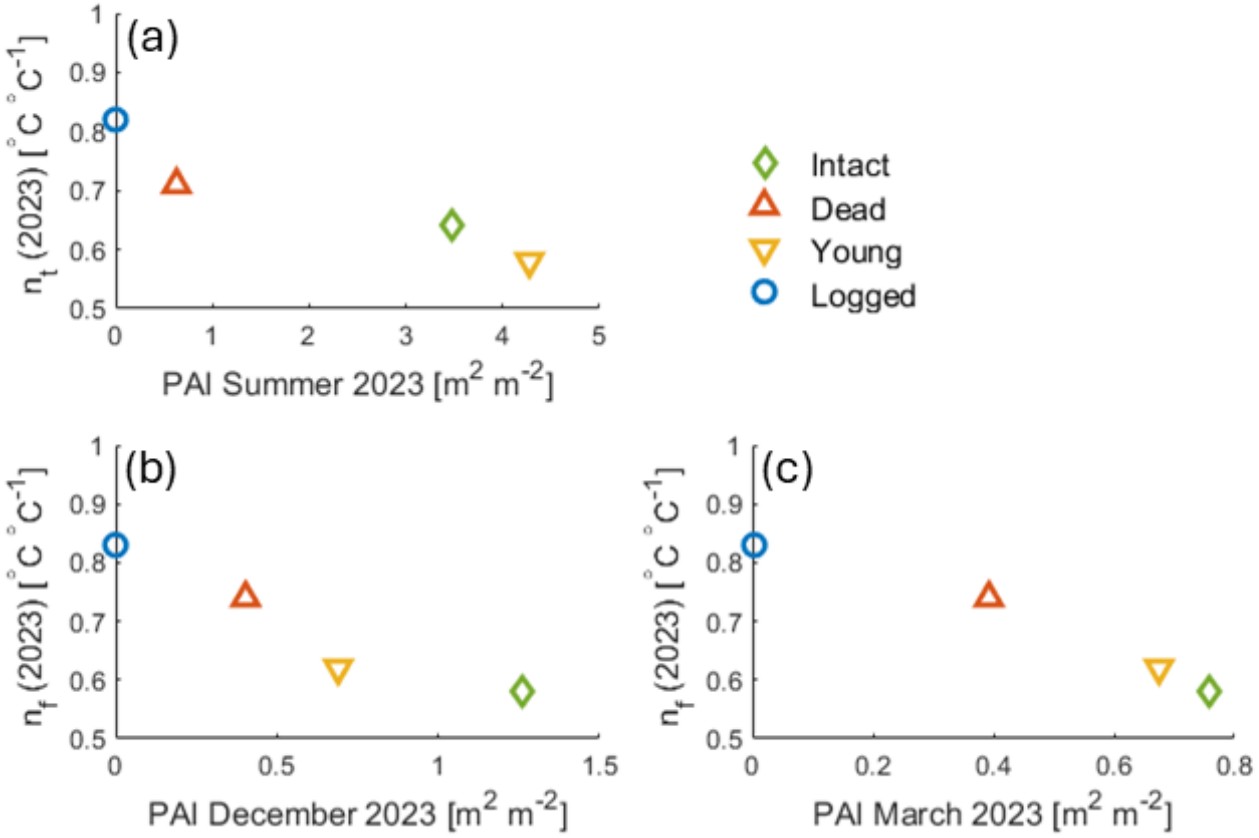

**Figure 6: (a) Comparison of summertime PAI$_{canopy}$ (19. August 2023) and the scale factor for thawing ($n_t$) over the hydrological year 2023. (b) & (c) Comparison of wintertime PAI$_{canopy}$ (21. December 2022 and 6. March 2023, respectively) and the scale factor for freezing ($n_f$) over the hydrological year 2023.**

### 3.3 Surface cover

Our observations of surface conditions reveal some general characteristics of the vegetation and snow cover for the different forest states. Snow surveys in December 2022 and March 2023, show snow depths of circa 15-20 cm for all forest states at both times, and no systematic differences in average snow depths between the field visits (Fig. C1). However, we note a substantial increase in livestock disturbance (trampling) to the snow surface from early to late winter (Table C2). While only scattered animal tracks were observed at all sites in December, the snow cover in the dead and logged forest had been substantially altered by animals by March, with more than half of the snow surface visually disturbed by animal activity. In contrast, we observed only scattered animal tracks in the intact forest throughout winter. In the stands of young regrowth our observations diverge, with extensive trampling observed in the aspen stand (YOUNG_1) in March, while we note only




scattered trampling within the larch stand (YOUNG_2). Comparing the average snow depths and the scale factor for freezing for the different forest states reveals no clear relationship (Fig. C2).

Surveys of surface cover show how floor vegetation and surface organic layers vary between the forest states (Table 3). In the intact forest we find floor vegetation of 10-20 cm height underlain by a thin layer of needle litter in both summer and winter. In the stands of young regrowth we observe only sparse floor vegetation and litter layers, while the dead forest features the highest floor vegetation of circa 20-25 cm (Fig. 2b) as well as the thickest litter layers in winter. In the logged forest we find a continuous, 5-15 cm high, grass cover in summer (Fig. 2a), except for at "LOGGED_2" where scattered
shrubs yield somewhat higher vegetation heights (Fig. 2c). Notably, we find surface organic layers at all sites in winter, but at most sites these are thinner or absent in summer (Table 3).

**Table 3: Observed surface cover from the visits in 6. March 2023, 19. August 2023 and 17. September 2024. Note that the surface organic layer was only qualitatively assessed in summer 2024. n.m.: not measured.**

| Field site | Floor vegetation | | Surface organic layers | | |
| --- | --- | --- | --- | --- | --- |
| | 2023 summer | 2024 summer | 2023 winter | 2023 summer | 2024 summer |
| INTACT_1 | n.m. | n.m. | 1 cm, litter | 4 cm, litter | Needle litter |
| INTACT_2 | 10-20 cm, grass | n.m. | 1-2 cm, litter | 1 cm, litter | Needle litter |
| DEAD_1 | 25 cm, grass | 20 cm, grass | 5 cm, grass litter | 0 cm | Grass litter |
| DEAD_2 | 25-50 cm, grass | 20 cm, grass | 5 cm, grass litter | 0 cm | Grass litter |
| DEAD_3 | 25 cm, grass | 15 cm, grass | 2 cm, litter/moss | 1 cm, moss | Litter and moss |
| YOUNG_1 | 3-4 m, Aspen saplings | 0 cm, grass | 2 cm, litter | 0 cm | Leaf litter |
| YOUNG_2 | 2-4 m, Larch saplings | n.m. | 2 cm, litter | 0 cm | Needle litter |
| LOGGED_1 | 15 cm, grass | 15 cm, grass | 5 cm, litter/moss | 0 cm, 10-20% bare soil | Negligible |
| LOGGED_2 | 20-30 cm, grass/shrubs | 40 cm, grass/shrubs | 2 cm, grass litter | 0 cm, 10-20% bare soil | Negligible |
| LOGGED_3 | 5-15 cm, grass | 15 cm, grass | 1 cm, grass litter | 0 cm | Negligible |

## 4 Discussion

### 4.1 Novel aspects and limitations

This study presents a novel dataset of GSTs, vegetation density and surface conditions that show how forest disturbances manifest in the forest-steppe ecotone in central Mongolia. Our study area in the southern Khentii mountain range is located near the southern border of both the Eurasian permafrost and boreal forest areas. The traditional lifestyle in this region is
closely intertwined with the existence of forest and permafrost, which provide firewood and seasonal pastures, as well as sustaining local streamflow as source of drinking water (Iijima et al., 2012; Karthe et al., 2013; Lkhagvadorj et al., 2013b, a). Research on interactions between ecosystems and ground thermal regime in this region have focused on the differences



between north-facing, permafrost underlain forest and the south-facing, permafrost-free steppe (e.g. Dashtseren et al., 2014; Ishikawa et al., 2005, 2018; Temuujin et al., 2024). Only few studies have investigated the impact of forest
disturbance on ground temperatures in this region, which can have strong implications for permafrost presence and forest succession (Klinge et al., 2021).

Previous studies from the forest-steppe ecotone of central Mongolia have shown a strong dampening of the annual GST cycle and a lowering of MAGSTs associated with forest cover (Dashtseren et al., 2014; Temuujin et al., 2024). However,
the ecosystem gradients in these studies occur across substantial differences in topography, and the direct effect of forests on GSTs remains unclear. To untangle the impacts of terrain and ecosystems on ground thermal regime, our study specifically targets differences in forest states across sites with only small differences in topographic characteristics (Table 1). Overall, we find somewhat lower MAGSTs at disturbed sites than in intact forest, while the annual GSTs range is larger at sites where forest disturbances have reduced the canopy cover (Fig. 3).


GSTs in our study area are likely also influenced by environmental factors other than forest state, such as variation in topography and floor vegetation. For example, the sites of dead forest and stands of young regrowth are located at roughly 50-100 m lower elevation and 2-3° steeper slopes than the intact and logged forest sites (Table 1). Such an elevation difference can entail up to 1°C higher air temperatures in the lower locations if assuming a dry adiabatic lapse rate
(9.8°C/km; Hartmann, D. L. (2016)). The somewhat steeper north-facing slopes in the dead forest and stands of young regrowth can reduce the direct solar radiation at these sites, an effect that is strongest during low solar angles in winter. However, we do not observe generally warmer summertime GSTs (nor colder wintertime GSTs) at the dead forest and stands of young regrowth compared to the intact and logged forest (Fig. 3). Furthermore, we find variations in floor vegetation within the forest states that potentially affect the measured GSTs. The largest variations in floor vegetation and
vegetation density are found in the logged forest, where differences in vegetation height of 25 cm and $PAI_{total}$ of 1.59 are accompanied by a spread in summertime GSTs of 1.8 and 2.5°C in 2023 and 2024, respectively. However, we find that despite the variations within the logged forest, LOGGED_1, LOGGED_2 and LOGGED_3 still have the lowest $PAI_{total}$ and highest summertime GSTs of all field sites. This indicates that the impact of variations in topography among the forest states, and vegetation cover within the forest states, is small compared to the control of forest states on GSTs.


In this study we describe the vegetation density using the plant area index (PAI), which despite its broad use has inherent limitations. The PAI was chosen as it gives a single, quantitative vegetation metric per site/field visit, and easily can be estimated using non-destructive optical methods and off-the-shelf software that accounts for relevant effects such as local slope and canopy clumping (Sect. 2.3). As the PAI includes all plant material, it is a useful measure of the canopy-
atmosphere interface and a suitable metric for e.g. overall light transmission in forests. However, factors such as exposure





and sky conditions influence the estimation of PAI (Chen et al., 1991), which can have contributed to the drop of 0.5 in PAI$_{canopy}$ in the intact forest in winter (Fig. 5). Furthermore, the PAI has limited application in fields such as hydrology and ecology, as processes such as transpiration and photosynthesis occur only through the leaf-/needle-area (Bréda, 2003), while accurate modelling of sub-canopy radiation relates more to the sky-view fraction and the transmissivity of direct solar

radiation (Jonas et al., 2020). In principle, these vegetation metrics can be derived from the hemispherical images using established methodologies, e.g. the leaf area index (LAI) of the canopy can be found as the difference in PAI$_{canopy}$ between summer and winter.

## 4.2 Factors impacting ground surface temperature dynamics in disturbed forests

A key effect of forest canopies is the attenuation of solar radiation, which reduces the available energy at the ground surface

(e.g. Mahat and Tarboton, 2012; Stuenzi et al., 2021b). The shading effect scales with canopy density, for example, Dashtseren et al. (2014) found that for a site near our study area only 14-22% of the incoming solar radiation penetrated the forest canopy in summer, while corresponding values for the leaf-less season where 63-67%. Overall, disturbances that remove or reduce the canopy density will increase the solar radiation available at the ground surface which can give higher GSTs, an effect which is strongest during high insolation in summer. This is in line with our observations, where we find  the

highest summertime GSTs at the forest states with the lowest canopy densities (Fig. 4), as well as the strong relation between summertime PAI$_{canopy}$ and the scale factor for thawing (Eq. 3, Fig. 6).

Forest canopies can also intercept substantial amounts of solar radiation in winter, which affects the surface energy balance at forested sites. In the forest-steppe ecotone, forests consist mainly of deciduous needle-leaves and are located on north-

facing slopes (Schlütz et al., 2008; Tsogtbaatar, 2004). During winter, when sun angles are lower and forest canopies have shed their needles, incoming solar radiation has both a longer travel distance through the canopy while also being able to penetrate deeper into the canopy. For similar latitudes as our sites, Webster et al. (2015) showed that larch tree trunks in winter are warmer than the surrounding air, emitting substantial amounts of longwave radiation to the snow surface. At times when absorption of solar radiation is limited by highly reflective snow cover, sites with dark tree trunks can thus experience

increased energy input to the local surface. This tendency of more available energy at forested sites in winter agrees well with our findings of higher wintertime GSTs and scale factor for freezing (Eq. 2) at the sites with higher PAI$_{canopy}$ in winter (Fig. 6b & c).

The exchange of energy between the atmosphere and ground at forested sites in also modified by the presence of layers of

surface organics, which in our study area consist mostly of needle- and grass litter (Table 3). The low thermal conductivity of such layers efficiently insulates the soil below, which affects the ground thermal regime through dampening of the annual ground temperature range and maintaining shallow active layer thickness' (Beringer et al., 2001; Bonan and Shugart,



1989; Chang et al., 2015; Kasischke and Johnstone, 2005). For the forest-steppe ecotone, thick surface organic layers are observed in intact forests and at grassland sites where grazing is limited (Dashtseren et al., 2014; Ishikawa et al., 2005; Zweigel et al., 2024a) while they are shallow at sites with degraded forest or young regrowth (Klinge et al., 2021). In this study we measure GSTs below any layers for surface organics (Sect. 2.2), and the effects of such layers are contained in the GST records. Overall, we observer rather thin (0-5 cm) surface organic layers in our study area, where needle litter are found year-round in the intact forest, while grass- and leaf litter layers are found in winter only at the disturbed forest sites (Table 3). With no or negligible layers of surface organics in the dead and logged forest in summer, a higher ground heat flux is expected at these sites. This is in line with the higher summertime GSTs and scale factor for thawing at these disturbed forest sites (Figs. 3 & 6). For the stands of young regrowth the effect of thin or absent layers of surface organics in summer is likely compensated for by the pronounced shading from the dense canopy (Fig. 5). We note that all forest states featured similar layers of surface organics at the base of the snowpack in winter (Table 3), and these should thus not be driving differences in winter GSTs. Overall, differences in the thickness of the surface organic layer likely contribute to the differences in annual GST range between the forest states, which can give deeper active layers especially in the logged and dead forest.

An interesting finding is that dead forest has similar $PAI_{total}$ as the intact larch forest (Fig. 5b) but features higher summertime GSTs (Fig. 3). This GST difference is observed despite grassland vegetation (which dominate the dead forest) having generally higher summer albedo than coniferous forest (to which larches belong) (Bonan, 2016), indicating lower absorption of solar radiation in the dead forest. While the exact reasons for these differences are unclear, there are notable difference in vegetation structure between these sites which can have implications for the processes by which energy is transfers from the vegetation to its surroundings. In the intact forest the largest part of the vegetation density is in the tall tree canopy, while it is in the floor vegetation in the dead forest (Fig. 5b). This implies that the absorption of solar radiation (and consequent heating of the vegetation) occurs much closer to the ground surface in the dead forest than in the intact forest, resulting in a larger temperature gradient between the vegetation and ground surface and subsequently more efficient energy transfer at this site. Furthermore, the relatively open canopy air space in the intact forest, and the exposure to greater wind speed higher up, can allow the intact forest canopy to more efficiently shed heat to the atmosphere.

Our results contrast some of the previous research on the impact of forest disturbances on the ground thermal regime. Generally, forest disturbances have been shown to increase ground temperatures, potentially leading to permafrost degradation (e.g. Fedorov et al., 2017; Iwahana et al., 2005; Li et al., 2021; Yoshikawa et al., 2002). For the forest-steppe ecotone of central Mongolia, Klinge et al. (2021) and Kopp et al. (2014) found summertime GSTs at sites affected by wildfire that were 3.1°C to 3.9°C warmer compared to nearby intact forest. Year-round studies of wildfires and clear-cuttings found that reduced evapotranspiration following disturbances strongly increased soil moisture levels, which delayed freezing





and deepened the active layer (Fedorov et al., 2017; Iwahana et al., 2005; Yoshikawa et al., 2002). In our study area, we observe a similar increase in summertime GST at disturbed sites as Klinge et al. (2021) and Kopp et al. (2014), but this warming is offset by overall lower GSTs in winter, and we do not observe slower freezing at disturbed sites (Fig. 3b & B1). Overall, our two years of measurements show MAGSTs consistently 0.5°C lower at disturbed sites compared to intact forest

(Fig. 3a). Unlike the study sites of Yoshikawa et al. (2002), Iwahana et al. (2005) and Fedorov et al. (2017), our study area is located on discontinuous permafrost in sloping terrain, which could give a more efficient drainage of soil water and thus inhibit the buildup of soil moisture following disturbances. While we do not have observations at deeper ground layers, this demonstrates that forest disturbances do not necessary lead to warming of ground temperatures, and that their impact on ground thermal regime is influenced by the local hydrological regime.


Furthermore, differences in grassland vegetation impact GST dynamics, which can explain the differences observed in the dead and logged forest. These sites feature similar topographic characteristics (Table 1), but we observe lower $PAI_{total}$ and only negligible layers of surface organics in the logged forest (Fig. 5, Table 3). Such a sparser surface cover intensifies the exchange of energy between the atmosphere and ground surface (Zweigel et al., 2024a), which is in line with the higher

summertime GSTs and larger scale factor for thawing observed in the logged forest (Fig. 3 & 6a). It is likely that these differences in summertime surface cover are due to livestock grazing, as we observe several active herder camps on south-facing slopes in the immediate surroundings of our study area (< 2 km) and the logged forest is easily accessible from the south (Fig. 1a).

Livestock can also affect GST dynamics in the cold season, when snow cover can exert significant control on the ground thermal regime (Zhang, 2005). The snow cover can vary substantially between sites with different vegetation cover, which is due to a) snow capture, where snow preferentially accumulates at sites with higher and denser vegetation (Hiemstra et al., 2002; Sturm et al., 2001), and b) interception loss, where denser canopies intercept more snow (Fisher et al., 2016; Lundberg and Koivusalo, 2003; Yi et al., 2007). At our sites, we observe rather similar snow depths for the different forest

states (Fig. C1), and do not find a clear relationship between observed snow depth and $n_f$ (Fig. C2). Notably, the main difference in snow cover between the forest states is the degree of animal trampling, which is characterized as scattered in the intact forest and extensive in the dead and logged forest (Table C2). If trampling leads to compaction of the entire snowpack, this can substantially reduce the thermal insulation of snow and give lower wintertime GSTs (e.g. Zweigel et al., 2024a). For both the dead and logged forest, the temperature records show sudden drops in GST for single loggers in winter

(Fig. B2b, c). A similar drop in GSTs is also evident for the stand of young aspen, where we also note substantial trampling (Table C2). While we do not have direct observations linking these GST signals to trampling, the extent of trampling observed at the sites in the dead and logged forest suggest that livestock disturbances to the snow cover can contribute to the overall colder winter GSTs at these sites.





Overall, livestock activity influences surface conditions in our study area, and these must be considered when studying the differences in GST dynamics between the forest states. While the exact patterns of livestock activity are unclear, we note some general tendencies; in summer, the sparser vegetation cover in the logged forest suggests that these sites are preferentially grazed (Fig. 5b), while the more abundant trampling at the dead and logged forest and the stand of young aspen indicates larger livestock activity at these sites in winter (Table C2). The latter is likely linked to the abundance of

edible grass and leaf litter at these sites contrary to the needle litter that dominates both young and intact larch forests (Table 3). For a north-facing grassland site in central Mongolia, Zweigel et al. (2024a) found grazing and trampling by livestock to give 1.4°C warmer and 2.5°C colder GSTs in summer and winter, respectively, and an overall cooling of MAGST of 0.4°C. It is thus likely that livestock activity contributes to the larger annual GST cycle and lower MAGSTs in the logged and dead forest sites compared to the intact forest (Fig. 3). As these impacts are superimposed on the forest disturbances in our study

area, quantification of their contribution to the observed GST dynamics is not possible. However, livestock activity likely also affects surface conditions throughout the forest-steppe ecotone, as 70-80% of the Mongolian land area is used as rangeland (Angerer et al., 2008; Fernández-Giménez et al., 2018) and forests constitute important feeding grounds that are used especially in winter (Lkhagvadorj et al., 2013b, a).

### 4.3 Implications and future work

In the forest-steppe ecotone both permafrost and forest are close to their climatic limits and their existence is intimately linked (e.g. Dashtseren et al., 2014; Klinge et al., 2018). In its intact state, forest ecosystems provide shade and insulation which limit ground thawing in summer, while the presence of shallow permafrost elevates the water table and increases local soil moisture levels. The potential for these systems to reestablish themselves after disturbances has received little attention, except for the work by Klinge et al. (2021). For burnt and exploited forests in central Mongolia, they found an overall drying

of the soil and deepening of the active layer. Modelling efforts have since underpinned the impact removal of forest canopy and associated surface organic layers have on soil moisture and seasonal thaw depth (Zweigel et al., 2024b). Such ground conditions are unfavourable for tree saplings, and initial succession is limited to topographic formations promoting water availability (Klinge et al., 2021). We also observe this effect in our study area, where sparse regrowth is found across the parts facing directly north, whereas stands of dense young regrowth (e.g. YOUNG_1 and YOUNG_2) are limited to

topographic concavities (Fig. 1). However, these stands of young regrowth feature the coldest summertime GSTs and the lowest scale factor for thawing (Fig. 3, Table 2), indicating a limited active layer thickness and locally stable permafrost at these sites. Such stands of young regrowth might be essential for the gradual regeneration of forest on disturbed slopes, as they provide necessary soil moisture and seed reservoir (Klinge et al., 2021). Fully understanding and quantifying the role of permafrost-ecosystem interactions for regenerating/degenerating forests is however beyond the scope of our study at this

time and should be a topic for future research.



The observed variation in GST dynamics between the different forest states can impact the local ground thermal regime. Overall, we find about 0.5°C lower surface offset (Eq. 1) in the dead and logged forest and stands of young regrowth compared to the intact forest (Table 2), which at least partially is linked to snow trampling in winter (Sect. 4.2). This indicates a general cooling of the ground surface following forest disturbance. However, the dead and logged forest have a larger annual GST amplitude, which in turn can increase the thermal offset and give higher ground temperatures at depth (Smith and Riseborough, 1996). Due to the potentially opposing effects of surface- and thermal offset, the impact of dead or logged forest on deeper ground temperatures remains unclear.

The abundant growth of floor vegetation at disturbed forest sites might limit remote sensing efforts in assessing the spatial scale of forest disturbance. Most remotely sensed vegetation indexes (i.e. NDVI) are based on the ratio between different spectral bands and thus do not distinguish between vegetation in the canopy and at the forest floor (Glenn et al., 2008). At our sites, we observe abundant grass vegetation in disturbed sites, yielding rather similar $PAI_{total}$ for intact and disturbed forest (Fig. 5b). Also previous works have found common vegetation indexes to be insensitive to disturbances in larch forest, as loss of canopy cover is compensated for by increased vitality of floor vegetation (Bendavid et al., 2023; Loranty et al., 2018). Remote sensing products based on spectral reflectance might thus be inadequate for detecting or mapping the spatial extent of such forest disturbances.

This observed relationship between vegetation density and GSTs suggests that canopy effects need to be considered when simulating permafrost in forest ecosystems. In recent years, single- or multi-layer vegetation schemes that directly account for canopy effects have been integrated in existing permafrost models and applied at the point scale (e.g. Stuenzi et al., 2021; Zweigel et al., 2024b). However, the faster and simpler models used to simulate regional and global permafrost extent often lump the effects of vegetation, snow cover and surface organics into scale factors linking air and ground surface temperatures (e.g. Jafarov et al., 2012; Obu et al., 2019; Westermann et al., 2015). In the frequently used permafrost map by Obu et al. (2019), direct canopy effects are not considered in the parameterization of the scale factors for freezing and thawing, and this is pointed out as a main source of uncertainty in the representation of permafrost extent in forested regions. However, the clear relationship between canopy density and the scale factors for thawing and freezing found in this study (Fig. 6) shows that canopies have a strong control on GSTs throughout the year. At our study sites, the impact of canopies on freezing and thawing largely cancel each other out, and surface offsets vary only by 0.5°C with forest state (Table 2). However, the strength of canopy effects such as shading, interception and snow capture vary with geographic location, climate and forest type, and should be considered in future permafrost mapping efforts.



The observed GST dynamics in the logged and dead forest are likely to affect the soil moisture regime, with important implications for ecosystem function. Specifically, higher summertime GSTs and larger annual GST range are typically associated with a deepening of the active layer (Smith and Riseborough, 1996; Zhang et al., 1997), and greater active layer thicknesses have been observed at disturbed forest sites in central Mongolia (Klinge et al., 2021). As permafrost is largely impermeable (Walvoord and Kurylyk, 2016), a deeper active layer permits water infiltration to greater depths while higher summertime GSTs at the same time increase surface evaporation. Consequentially, near-surface moisture levels are reduced which limits water available for plant uptake. Moreover, meltwater from ground ice near the top of the permafrost can be crucial in sustaining boreal forests during dry years (Sugimoto et al., 2002). If forest disturbances lead to deepening of the active layer and a melt out of these ground ice layers, this important water reservoir is lost and any post-disturbance ecosystems will be more susceptible to droughts. Overall, forest disturbances can negatively affect both short- and long-term water availability, which has implications for ecosystem function and succession. Although investigating water availability is beyond the scope of this study, the similar terrain configurations among sites in the Bayanzurkh mountains suggest that it could be a well-suited location for future research on the impact of forest disturbances on soil moisture dynamics.

## 5 Conclusion

In this study we explore the relationship between ground surface temperatures (GST), vegetation cover and surface organic layers at intact and disturbed forest sites in the forest-steppe ecotone in central Mongolia. We present two years of daily GST measurements from sites with intact, dead, and logged forest, as well as from dense stands of young regrowth. The GST measurements are complemented by daily air temperature measurements, calculations of canopy and floor vegetation density and surveys of surface cover in summer and winter.

We find forest disturbances to strongly affect vegetation structure, with the dead and logged forest featuring lower canopy density, denser floor vegetation and lower accumulation of surface organic layers. Stands of young regrowth have the highest summertime canopy density and the sparsest floor vegetation, and feature only a negligible surface organic layer. The forest disturbance also induces differences in local livestock activity especially in winter when grass- and leaf litter available at disturbed sites is an important source of feed. Overall, there is a clear relationship between canopy density and winter and summer GSTs, where the annual GST amplitude increases at sites with sparser canopies. Compared to intact forest, monthly GSTs in the dead and logged forest are up to +1.8°C and +3.7°C higher in summer and up to -3.6°C and -5.1°C lower in winter, respectively. In stands of young regrowth, GSTs are generally lower and more dampened than in intact forest. We find summer and winter effects to largely offset each other, and that mean annual GSTs in the dead, logged and stands of young regrowth are 0.5°C lower than in the intact forest.





Our findings suggest that while forest disturbances strongly influence the GST dynamics, they do not lead to widespread

warming of the ground surface. Forest disturbances lead to a shift in vegetation density from the canopy to the forest floor, which can impede remote sensing of forest disturbances using well-established vegetation indices and induce patterns in livestock activity.

**Appendix A – Location of the field sites**

*Table A4: Coordinates of the field sites used in this study.*

| Logger | Latitude [°] | Longitude [°] | |
|---|---|---|---|
| INTACT_1 | 47.8334 | 107.2137 | 530 |
| DEAD_1 | 47.8348 | 107.2165 | |
| DEAD_2 | 47.8336 | 107.2175 | |
| YOUNG_1 | 47.8333 | 107.2179 | |
| YOUNG_2 | 47.8326 | 107.2190 | |
| DEAD_3 | 47.8326 | 107.2193 | 535 |
| LOGGED_1 | 47.8295 | 107.2202 | |
| LOGGED_2 | 47.8294 | 107.2206 | |
| LOGGED_3 | 47.8303 | 107.2174 | |
| INTACT_2 | 47.8324 | 107.2129 | |



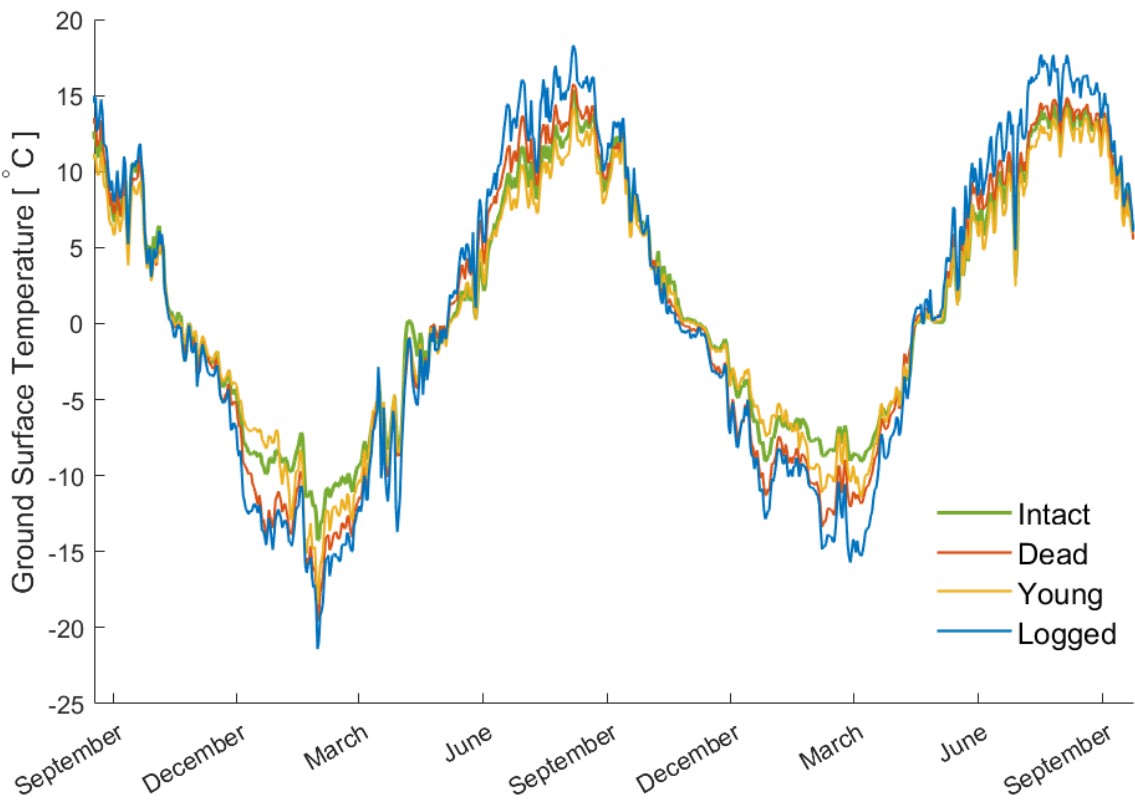

**Figure B1: Daily GSTs evolution for the different forest states.**







**Figure B2: Daily mean GST (black line) and GST range (coloured shading) for the 2-3 temperature measurements within the different forest states (Sect. 2.2). Note the abrupt increases in GST range in winter for some forest states, which is due to drops in GST for single loggers.**




## Appendix C – Detailed vegetation and snow observations

**Table C1: Calculated PAI for the different field sites. * PAI$_{canopy}$ values for logged sites in summer are not calculated directly but**
**are assumed to be equal to those calculated for winter (see Sect. 3.2).**

| Logger | Forest state | PAI$_{canopy}$ | | | PAI$_{total}$ |
|---|---|---|---|---|---|
| | | 22.12.2022 | 06.03.2023 | 19.08.2023 | 19.08.2023 |
| INTACT_1 | Intact | 1.15 | 0.77 | 3.60 | 4.20 |
| DEAD_1 | Dead | 0.30 | 0.32 | 0.34 | 4.94 |
| DEAD_2 | Dead | 0.40 | 0.41 | 0.40 | 4.51 |
| YOUNG_1 | Yong | 0.58 | 0.63 | 4.14 | 5.39 |
| YOUNG_2 | Young | 0.80 | 0.73 | 4.41 | 4.53 |
| DEAD_3 | Dead | 0.51 | 0.45 | 1.15 | 4.17 |
| LOGGED_1 | Logged | 0.00 | 0.00 | 0.00* | 2.45 |
| LOGGED_2 | Logged | 0.00 | 0.01 | 0.00* | 1.38 |
| LOGGED_3 | Logged | 0.00 | 0.00 | 0.00* | 0.86 |
| INTACT_2 | Intact | 1.37 | 0.75 | 3.35 | 4.94 |

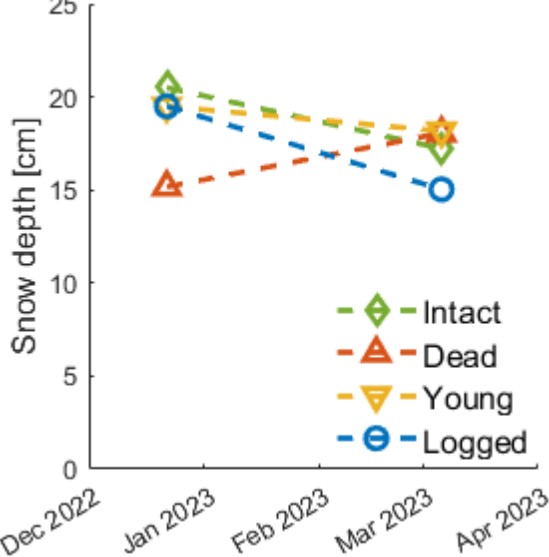

**Figure C1: Average snow depths measured in the immediate vicinity of the logger locations in winter 2023. Note that the data from
6. March 2023 is the mean of measurements at both visually undisturbed and disturbed locations.**




**Table C1: Observations of snow cover at the individual field sites. Note that the snow depths on 06.03.2023 are the mean values of several snow measurements. Trampling is characterized as "scattered" and "extensive" if respectively less than 10% or more than 50% of the snow surface are visually disturbed by livestock.**

| Logger | Forest state | 22.12.2022 | | 06.03.2023 | |
|---|---|---|---|---|---|
| | | Snow depth | Trampling | Snow depth | Trampling |
| INTACT_1 | Intact | 21 cm | Scattered | 18 cm | Scattered |
| DEAD_1 | Dead | 10 cm | Scattered | 19 cm | Extensive |
| DEAD_2 | Dead | 19 cm | Scattered | 16 cm | Extensive |
| YOUNG_1 | Yong | 21 cm | Scattered | 20 cm | Extensive |
| YOUNG_2 | Young | 18 cm | Scattered | 17 cm | Scattered |
| DEAD_3 | Dead | 16.5 cm | Scattered | 20 cm | Extensive |
| LOGGED_1 | Logged | 20 cm | Scattered | 15 cm | Extensive |
| LOGGED_2 | Logged | 21.5 cm | Scattered | 16 cm | Extensive |
| LOGGED_3 | Logged | 17 cm | Scattered | 14 cm | Extensive |
| INTACT_2 | Intact | 20 cm | Scattered | 16 cm | Scattered |


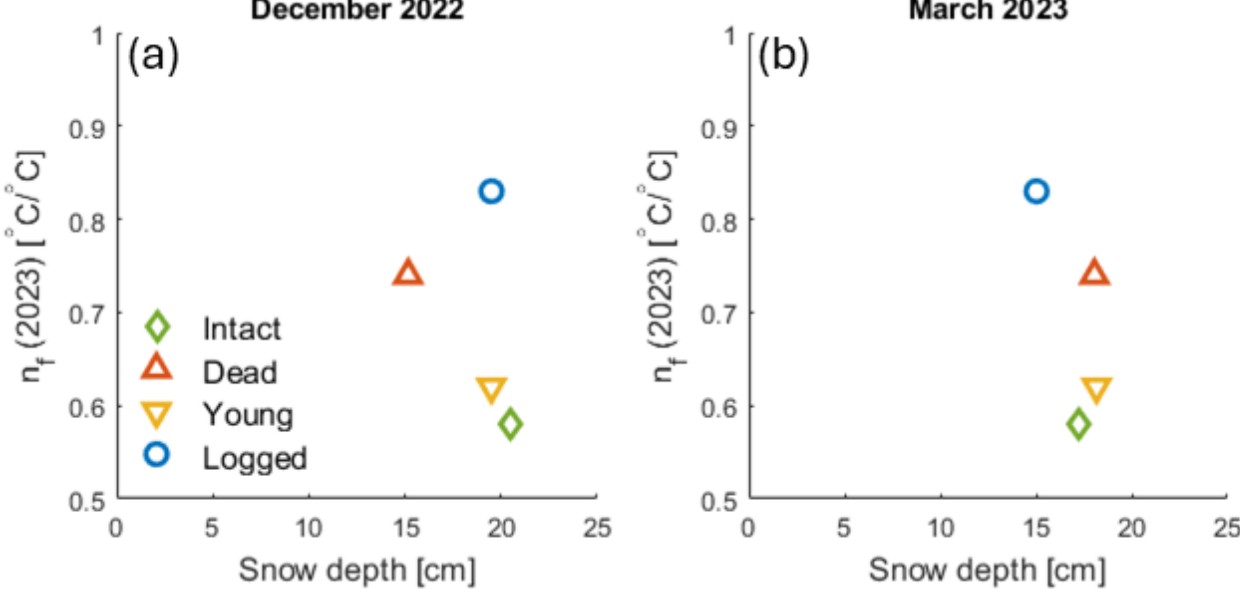

**Figure C2: Comparison of average snow depth and the scale factor for freezing ($n_f$) over the hydrological year 2023.**



**Data availability**

The ground surface temperature data, hemispherical images, snow-, surface cover-, and vegetation surveys are available from Zweigel, R. (2025).

**Author Contribution**

RBZ and SW conceptualized the study and collected the field data together with AD, TK and CW. RBZ led the formal analysis and investigation and wrote the initial draft with contributions from all co-authors. AD and SW administrated the 570 project, and HL and SW acquired funding and contributed with supervision.

**Competing interests**

One of the co-authors is on the editorial board of The Cryosphere.

**Acknowledgments**

We thank the editor and reviewers for taking the time to evaluate this manuscript.

**Financial support**

This research has been supported by the Research Council of Norway (Permafrost4Life (grant no. 301639), PRISM (grant no. 309625)) and the European Space Agency CCI+ Permafrost (grant no. 4000123681/18/I-NB).

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
