# Peer review of "Forest disturbances and their impact on ground surface temperatures in permafrost-underlain forest in central Mongolia"

_EGUsphere, 2025_

## Author Comment (AC1)

In this author reply, we have carefully gone through the comments and suggestions made by the referee, which are shown in *blue italics*. We provide our response in normal font whereas our suggestions for the revised manuscript are shown as **bold text**. Line numbers refer to the original submitted manuscript.

Ground surface temperature (GST) is a crucial factor for permafrost research and predictive mapping. However, the current study requires substantial improvements in the following aspects:

We thank the referee for the time to review the manuscript and for their acknowledgment of the importance of the research topic and the presented dataset. A main goal of this manuscript is indeed to bring forward a novel GST dataset, and interpretation thereof, from a region that is underrepresented in the literature and where permafrost and boreal forest exist close to their climatic limits. We agree with the referee that modifications are warranted to improve the quality of the manuscript, including more detailed description of permafrost characteristics in this region and more quantitative analysis of the factors regulating GSTs.

**Mechanistic Analysis**

It is necessary to establish quantitative relationships between various influencing factors and GST, and to calculate the contribution rate of each factor.

Simply determining the dominant influencing factors through qualitative discussion is scientifically inadequate.

We agree with the referee that a more quantitative assessment of the factors impacting GSTs is warranted. In response to this comment, we have conducted a correlation analysis between different annual and seasonal GST metrics and various variables related to vegetation, surface cover and topography. By incorporating such an analysis in our study, we can evaluate the strength of any (linear) relationship between the included variables, which will form the basis for our discussion of the factors controlling GST dynamics across the different forest states.

The correlation analysis reveals several important aspects of the relationships between GSTs and environmental variables. Notably, the analysis shows strong correlations between seasonal GSTs and the related temperature metrics and vegetation densities (especially PAIcanopy), while differences in surface cover (snow and organic layers) and topographic characteristics have no strong correlation to GSTs. This suggest that the seasonal GST dynamics in our field area are primarily linked to the effect that forest disturbances have on canopy densities.

These findings have helped us refine the study, including a strengthened foundation for our discussion of vegetation densities and a de-empathising of the role of surface cover in shaping seasonal GST. In the revised manuscript, we will including the main findings of the correlation analysis as the first result chapter (Sect. 3.1), while the relationships for the strongest correlating factors (e.g. between PAIcanoy and  $n_f/n_t$ ; Eqs. 2 & 3) are shown in the respective results chapters. The discussion will be revised in light of these findings, and the full correlation analysis will be shown in appendix D.

**3.1 Correlation between GSTs and environmental factors**

We conduct a correlation analysis to quantitatively assess the presence of strong relationships between GSTs and environmental factors across the disturbance gradient targeted by the field sites. In this

analysis, we calculate the correlation between annual and seasonal GSTs metrics and different environmental variables related to vegetation density, surface cover and terrain (Appendix D). From Table D1 it is evident that the seasonal GST metrics are strongly correlated to vegetation, while there are no strong correlations between MAGST and any of the environmental factors. Specifically, the correlation analysis shows that higher PAIcanopy is associated to higher summertime GSTs and lower wintertime GSTs, as well as a smaller seasonal GST range. Interestingly, we do not find any strong correlations between GSTs and the thickness of surface layers of snow or organics. Furthermore, the analysis shows no strong correlations between GSTs and the different terrain metrics, which is expected as the field sites are specifically chosen to have similar terrain characteristics (Sect. 2.1). The lack of strong correlations between terrain metrics and GSTs thus suggests that the remaining terrain variations among the field sites do not substantially influence GSTs. The analysis also shows strong internal correlation within several of the variable types, for example among the different terrain metrics (Table D1). Overall, the correlation analysis reveals a strong relationship between seasonal GSTs and vegetation densities among the different forest states at Bayanzurkh.

Line 344: This is in line with the correlation analysis, which shows a strong relationship between vegetation density and summertime GSTs and the scale factor for thawing (Table D1), with the lowest summertime GSTs and  $n_f$  found at sites with higher summertime PAIcanopy (Figs. 4 and 6, Eq. 3).

Line 354: This tendency of more available energy at forested sites in winter agrees well with our findings of higher wintertime GST and lower scale factor for freezing (Eq. 2) at sites with denser canopies (Figs. 6b, c) and the strong positive correlations between PAIcanopy and wintertime GSTs (Table D1).

Line 369: However, the correlation analysis does not indicate any strong relationships between the surface organic layer thickness and seasonal GSTs nor the scale factors for freezing and thawing. This suggests that variations in surface organic layer thickness among the forest states are not the dominant factor influencing GST dynamics in our study area.

Line 399: The correlation analysis also does not show any strong correlations between MAGST and the variations in surface and vegetation cover across the different forest states (Table D1).

Line 419: Interestingly, we observe rather similar snow depths for the different forest states in Bayanzurkh (Fig. C1) and find no clear relationship between observed snow depth and  $n_f$  (Fig. C2) nor a strong correlation between snow depths and wintertime GSTs (Table D1). This lack of systematic differences in snow cover between the forest states could be linked to limited wind redistribution of snow due to the overall sheltered location of the field sites (Sect. 2.1) and the generally low wintertime windspeeds in central Mongolia (Hong et al., 2023), or the deciduous nature of the forest in our field area, which limits snow interception.

**Appendix D – Correlation analysis**

To determine the dominant factors influencing GST dynamics in our study area we conduct a correlation analysis, which provides a quantitative measure of the strength of any linear relationships between the included variables. This analysis seeks to relate temperature indices (the target variables) to potential explanatory variables related to different environmental factors. The target variables include annual and seasonal GSTs, the range of seasonal GSTs, and the scale factors for freezing and thawing. As potential explanatory variables we incorporate the measured vegetation densities (including the range of PAIcanopy), the thickness of surface organic layers and snow cover, and the various terrain-related metrics presented in Table 1. Due to the limited sample size available (10 field sites), we only consider strong correlations (above 0.7 or below -0.7) in our analysis.

Table D1: Correlation between temperature indices and various metrics related to vegetation, surface cover and terrain for the hydrological year 2023. Here, the temperature indices are considered target variables (italic font), while the different vegetation, surface and terrain metrics (normal font) are considered possible explanatory variables. Correlations above or below 0.7 and -0.7, respectively, are considered strong correlations, and these are indicated by bold font and coloured shading (green for positive correlations and red for negative correlations).  $GST_{range}$  refers to the difference between summer and wintertime GSTs, while  $d_{SOL}$  and  $d_{snow}$  are the depth of the surface organic layer and snow cover, respectively. For variables that relate to a specific time or period, this is indicated in brackets.  $S_{in}$  is considered a terrain metric as the field sites are situated in close spatial proximity, so that differences in available solar radiation are induced by terrain variations between the sites.

|                  |                      | Temperature indices |                 |                 |               |       |       | Vegetation density                  |                                  |                                   |                                        |                                  | Surface cover                      |                          |                             |                          | Terrain metrics |       |        |       |       |                 |                 |
|------------------|----------------------|---------------------|-----------------|-----------------|---------------|-------|-------|-------------------------------------|----------------------------------|-----------------------------------|----------------------------------------|----------------------------------|------------------------------------|--------------------------|-----------------------------|--------------------------|-----------------|-------|--------|-------|-------|-----------------|-----------------|
|                  |                      | MAGST               | GST
(Winter) | GST
(Summer) | $GST_{range}$ | $n_f$ | $n_t$ | PAI canopy
(December) | PAI canopy
(March) | PAI canopy
(August) | PAI canopy
(Aug. – Mar.) | PAI total
(August) | d SOL
(March) | d SOL
(August) | d snow
(December) | d snow
(March) | Elevation       | Slope | Aspect | SVF   | Sin   | Sin
(Winter) | Sin
(Summer) |
| Target variables | MAGST                | 1.00                | 0.50            | 0.17            | -0.21         | -0.63 | 0.30  | 0.23                                | 0.14                             | 0.09                              | 0.08                                   | 0.25                             | 0.51                               | 0.22                     | 0.38                        | -0.16                    | 0.20            | 0.16  | -0.61  | -0.06 | 0.06  | 0.08            | 0.07            |
|                  | GST
(Winter)      | 0.50                | 1.00            | -0.74           | -0.94         | -0.97 | -0.66 | 0.84                                | 0.85                             | 0.83                              | 0.80                                   | 0.70                             | -0.14                              | 0.46                     | 0.43                        | 0.41                     | 0.05            | 0.61  | -0.32  | -0.41 | -0.60 | -0.57           | -0.59           |
|                  | GST
(Summer)      | 0.17                | -0.74           | 1.00            | 0.92          | 0.65  | 0.99  | -0.77                               | -0.89                            | -0.86                             | -0.83                                  | -0.74                            | 0.39                               | -0.24                    | -0.05                       | -0.60                    | 0.21            | -0.64 | 0.02   | 0.54  | 0.69  | 0.65            | 0.69            |
|                  | GST range | -0.21               | -0.94           | 0.92            | 1.00          | 0.88  | 0.87  | -0.86                               | -0.93                            | -0.90                             | -0.87                                  | -0.77                            | 0.27                               | -0.39                    | -0.27                       | -0.53                    | 0.07            | -0.67 | 0.19   | 0.50  | 0.68  | 0.65            | 0.68            |
|                  | $n_f$                | -0.63               | -0.97           | 0.65            | 0.88          | 1.00  | 0.55  | -0.77                               | -0.79                            | -0.74                             | -0.72                                  | -0.75                            | -0.06                              | -0.37                    | -0.38                       | -0.37                    | 0.00            | -0.63 | 0.49   | 0.46  | 0.50  | 0.46            | 0.49            |
| _                | $n_t$                | 0.30                | -0.66           | 0.99            | 0.87          | 0.55  | 1.00  | -0.71                               | -0.83                            | -0.82                             | -0.79                                  | -0.65                            | 0.48                               | -0.22                    | -0.06                       | -0.63                    | 0.21            | -0.60 | -0.06  | 0.50  | 0.68  | 0.65            | 0.68            |

|                       | PAI canopy
(December) | 0.23  | 0.84  | -0.77 | -0.86 | -0.77 | -0.71 | 1.00  | 0.93  | 0.81  | 0.77  | 0.68  | -0.37 | 0.62  | 0.21  | 0.34  | 0.29  | 0.36  | -0.03 | -0.18 | -0.47 | -0.46 | -0.45 |
|-----------------------|-------------------------------------|-------|-------|-------|-------|-------|-------|-------|-------|-------|-------|-------|-------|-------|-------|-------|-------|-------|-------|-------|-------|-------|-------|
|                       | PAI canopy
(March)    | 0.14  | 0.85  | -0.89 | -0.93 | -0.79 | -0.83 | 0.93  | 1.00  | 0.89  | 0.85  | 0.83  | -0.31 | 0.51  | 0.13  | 0.55  | -0.05 | 0.54  | -0.04 | -0.46 | -0.61 | -0.58 | -0.60 |
|                       | PAI canopy
(August)   | 0.09  | 0.83  | -0.86 | -0.90 | -0.74 | -0.82 | 0.81  | 0.89  | 1.00  | 1.00  | 0.64  | -0.49 | 0.39  | 0.33  | 0.47  | 0.02  | 0.39  | 0.05  | -0.29 | -0.51 | -0.51 | -0.52 |
|                       | PAI canopy
(AugMar.)  | 0.08  | 0.80  | -0.83 | -0.87 | -0.72 | -0.79 | 0.77  | 0.85  | 1.00  | 1.00  | 0.59  | -0.51 | 0.36  | 0.36  | 0.45  | 0.03  | 0.35  | 0.07  | -0.25 | -0.48 | -0.48 | -0.49 |
|                       | PAI total
(August)    | 0.25  | 0.70  | -0.74 | -0.77 | -0.75 | -0.65 | 0.68  | 0.83  | 0.64  | 0.59  | 1.00  | 0.19  | 0.19  | -0.18 | 0.68  | -0.40 | 0.77  | -0.38 | -0.77 | -0.58 | -0.50 | -0.58 |
|                       | d SOL
(March)  | 0.51  | -0.14 | 0.39  | 0.27  | -0.06 | 0.48  | -0.37 | -0.31 | -0.49 | -0.51 | 0.19  | 1.00  | -0.44 | -0.39 | -0.14 | -0.42 | 0.32  | -0.61 | -0.41 | 0.08  | 0.16  | 0.08  |
| Expl                  | d SOL
(August)        | 0.22  | 0.46  | -0.24 | -0.39 | -0.37 | -0.22 | 0.62  | 0.51  | 0.39  | 0.36  | 0.19  | -0.44 | 1.00  | 0.26  | 0.26  | 0.44  | -0.07 | 0.04  | 0.18  | -0.07 | -0.12 | -0.07 |
| Explanatory variables | d snow
(December)     | 0.38  | 0.43  | -0.05 | -0.27 | -0.38 | -0.06 | 0.21  | 0.13  | 0.33  | 0.36  | -0.18 | -0.39 | 0.26  | 1.00  | -0.13 | 0.38  | -0.17 | -0.04 | 0.28  | 0.02  | -0.05 | 0.02  |
| y varia               | d snow
(March)        | -0.16 | 0.41  | -0.60 | -0.53 | -0.37 | -0.63 | 0.34  | 0.55  | 0.47  | 0.45  | 0.68  | -0.14 | 0.26  | -0.13 | 1.00  | -0.55 | 0.68  | -0.15 | -0.72 | -0.58 | -0.54 | -0.61 |
| bles                  | Elevation                           | 0.20  | 0.05  | 0.21  | 0.07  | 0.00  | 0.21  | 0.29  | -0.05 | 0.02  | 0.03  | -0.40 | -0.42 | 0.44  | 0.38  | -0.55 | 1.00  | -0.61 | 0.15  | 0.85  | 0.45  | 0.39  | 0.48  |
|                       | Slope                               | 0.16  | 0.61  | -0.64 | -0.67 | -0.63 | -0.60 | 0.36  | 0.54  | 0.39  | 0.35  | 0.77  | 0.32  | -0.07 | -0.17 | 0.68  | -0.61 | 1.00  | -0.42 | -0.91 | -0.81 | -0.73 | -0.81 |
|                       | Aspect                              | -0.61 | -0.32 | 0.02  | 0.19  | 0.49  | -0.06 | -0.03 | -0.04 | 0.05  | 0.07  | -0.38 | -0.61 | 0.04  | -0.04 | -0.15 | 0.15  | -0.42 | 1.00  | 0.35  | -0.14 | -0.25 | -0.14 |
|                       | SVF                                 | -0.06 | -0.41 | 0.54  | 0.50  | 0.46  | 0.50  | -0.18 | -0.46 | -0.29 | -0.25 | -0.77 | -0.41 | 0.18  | 0.28  | -0.72 | 0.85  | -0.91 | 0.35  | 1.00  | 0.71  | 0.63  | 0.72  |
|                       | Sin                                 | 0.06  | -0.60 | 0.69  | 0.68  | 0.50  | 0.68  | -0.47 | -0.61 | -0.51 | -0.48 | -0.58 | 0.08  | -0.07 | 0.02  | -0.58 | 0.45  | -0.81 | -0.14 | 0.71  | 1.00  | 0.99  | 1.00  |
|                       | Sin
(Winter)                     | 0.08  | -0.57 | 0.65  | 0.65  | 0.46  | 0.65  | -0.46 | -0.58 | -0.51 | -0.48 | -0.50 | 0.16  | -0.12 | -0.05 | -0.54 | 0.39  | -0.73 | -0.25 | 0.63  | 0.99  | 1.00  | 0.99  |
|                       | Sin
(Summer)                     | 0,07  | -0,59 | 0,69  | 0,68  | 0,49  | 0,68  | -0,45 | -0,60 | -0,52 | -0,49 | -0,58 | 0,08  | -0,07 | 0,02  | -0,61 | 0,48  | -0,81 | -0,14 | 0,72  | 1,00  | 0,99  | 1,00  |

**Permafrost Context**

GST studies alone cannot confirm whether the study area is located in a permafrost zone.

We recommend: Incorporating existing permafrost data in the study area description. Providing a map clearly indicating the study area's position within the permafrost region

We agree with the referee that the permafrost context should be included in the description of the study area. Indeed, extensive permafrost research has been conducted at the Terelj station located circa 20 km northeast of Bayanzurkh. In Terelj, atmospheric, surface and ground variables have been measured for two decades at an intact forest site with similar slope, aspect and elevation as our field sites in Bayanzurkh. Furthermore, we agree that the permafrost zonation of the study area and its surroundings is relevant context. Referee #2 had a similar suggestion, and we our suggested amendments address both these comments.

For the revised manuscript we will include the following expansion of section 2.1 with a description of permafrost conditions and revised Fig. 1 including permafrost zonation:

Line 75: This region of Mongolia is located at the southern margin of the Eurasian permafrost extent, with a transition in the study area from continuous permafrost in high-elevation areas in the northeast to sporadic permafrost in lower lying areas in the southwest. The study area is located in the discontinuous permafrost zone (Fig. 1b), where permafrost and forest cover are co-located on north-facing slopes, while south-facing slopes feature permafrost-free steppe (Temuujin et al., 2024). Climatic, permafrost and subsurface conditions have been studied for two decades around the Terelj station (Fig. 1b), which features similar topographic, climatic and vegetation characteristics as our study area (Dashtseren et al., 2014; Ishikawa et al., 2005; Temuujin et al., 2024). Here, borehole data below a mature larch forest shows wintertime and summertime GSTs of -9.4°C to-11.2°C and 9.0°C to 10.2°C, respectively, and permafrost with an active layer of 2.4 - 2.7 m (Dashtseren et al., 2014). Furthermore, Temuujin et al. (2024) studied the relationship between topographic characteristics, vegetation cover and GSTs in Terelj using spatially distributed measurements and found lower MAGSTs (-1°C to 1°C) below forest and on sites with limited available solar radiation.

Figure 1: Overview maps of the study area. a) Shows the main landcover topographic characteristics of the study area in the Bayanzurkh mountains, as well as the location of the GST loggers and the air temperature measurement. The coloured markers for the field sites (GST loggers) are grouped by forest states. Background imagery from 10. September 2022 (Esri, 2023) and contour lines (100 and 25 m) derived from the SRTM 30 m Digital Elevation Model (NASA JPL, 2013). b) Location of the study area and places mentioned in the text within the Ulaanbaatar area, showing the permafrost zonation by Obu et al. (2019) (1x1 km resolution) overlaid a hillshade of the terrain (NASA JPL, 2013).

**Line 40: Such disruptions? The above text does not mention such disruptions.**

We thank the referee for this remark. "Such disruptions" refers to "Major changes in forest structure and functioning" in the previous sentence. In the revised manuscript, will reword this for clarity:

Line 38: Disturbances to the forest cover impact its structure and function, which has implications for the ground hydrological- and thermal regime, and potentially can impact permafrost stability (Stuenzi et al., 2021a, 2022). Such disturbances to the forest cover can

be biotic, for example insect infestations or diseases, or abiotic like wildfires, logging or windthrow, and we referred to them collectively as "forest disturbances".

Lines 45-49: Many studies have also examined the changes in winter in the permafrost region.

We agree with the referee that this phrasing is inaccurate. We have reviewed the presented literature and found that albeit focusing on summer aspects, Fedorov et al. (2017) and Yoshikawa et al., 2002 also includes wintertime temperatures and snow depths. In the revised manuscript we will include the following revised lines:

Line 46: However, most studies on forest disturbances in discontinuous permafrost in Mongolia focus on summer conditions (Klinge et al., 2021; Kopp et al., 2014), while year-round studies are limited to flat areas or overall different climates (Fedorov et al., 2017; Yoshikawa et al., 2002). For this reason, the impact of forest disturbances on the seasonal and annual ground temperatures at sites where hydrological and thermal conditions are strongly influenced by local terrain remains unclear.

For Introduction: It is essential to highlight the significance of ground surface temperatures (GSTs) research in the introduction, particularly in the context of permafrost. Studying GST can reveal its connection with permafrost, such as whether it can reflect the dynamic changes in permafrost conditions."

We agree that more background on the role of GST measurements in permafrost research is warranted. We thus will include the following paragraph in the revised manuscript:

Line 61: Ground surface temperature measurements are an increasingly valuable tool for research into permafrost dynamics. Unlike traditional permafrost borehole data, the acquisition of GST measurements is rather inexpensive and logistically simple, which facilitates spatially distributed observation of ground thermal dynamics. As GSTs are sampled at or near the ground surface, they include the influence of snow or vegetation layers, as well as any effects of variation in solar radiation caused by local terrain (e.g. Temuujin et al., 2024). Moreover, GST measurement can reveal dynamic changes in permafrost conditions as they respond quickly to shifts in climatic conditions or land use. While GSTs cannot directly reveal the presence or stability of permafrost, mean annual GSTs (MAGSTs) are linked to deeper ground temperatures through the thermal offset. Due to the generally higher thermal conductivity of frozen than thawed soil, the thermal offset is typically negative, which means permafrost can be expected also for somewhat positive MAGSTs (Smith and Riseborough, 2002). Moreover, GST measurements provide important validation data for permafrost modelling efforts (e.g. Schmidt et al., 2021; Zweigel et al., 2021 & 2024b).

*Line 101: What is A1?*

This is a typo that we will correct in the revised manuscript:

Line 100: Within the study area we select 10 field sites targeting the four distinct forest states found in the study area: intact forest, dead forest, logged forest and stands of young regrowth (Error! Reference source not found. & A1).

Figures 1 and 2: The general title for all figures should be provided first, followed by individual captions for each subplot.

We will included a general title in all figures that include multiple subplots:

Figure 2: Overview maps of the study area. (a) Shows the main landcover and topographic characteristics of the study area in the Bayanzurkh mountains, as well as the location of the GST loggers and the air temperature measurement. The coloured markers for the loggers are grouped by the apparent forest states. Background imagery from 10. September 2022 (Esri, 2023) and contour lines (100 and 25 m) derived from the SRTM 30 m Digital Elevation Model (NASA JPL, 2013). (b) Location of the study area and places mentioned in the text within the Ulaanbaatar area, showing the permafrost zonation by Obu et al. (2019).

Figure 3: Photographs showing the vegetation cover in the study area. (a) 25. June 2022: Looking north from the saddle point close to LOGGED\_3, showing the transition from logged and grazed areas in the foreground, to scattered, young regrowth mixed with dead tree trunks in the lower parts of the central valley. The intact and dead forest in the western valley is visible above the ridge in the background. (b) 19. August 2023: Dead forest and dense floor vegetation in the vicinity of DEAD\_1. (c) 19. August 2023: Logged forest at LOGGED\_2 with floor vegetation consisting of grasses and shrubs (knife handle 10 cm for scale).

Figure 4: Plot showing how GSTs vary with forest state in 2023 and 2024. (a) depicts mean annual GST while (b) shows seasonal GSTs. SON: September, October and November. DJF: December, January and February. MAM: March, April and May. JJA: June, July and August. Note the different scales on the temperature axes of (a) and (b). Daily mean GST and GST range for the different forest states are presented in Figs. B1 and B2.

Figure 5: Variations in vegetation densities between the forest states. (a) shows the seasonal evolution of PAIcanopy in 2023, while (b) compares PAIcanopy (open markers) and PAItotal (filled markers) on the 19th August 2023

Figure 6: Relationships between vegetation density and the scale factors for freezing and thawing. (a) Comparison of summertime  $PAl_{canopy}$  (19. August 2023) and the scale factor for thawing ( $n_t$ ) over the hydrological year 2023. (b) & (c) Comparison of wintertime  $PAl_{canopy}$  (21. December 2022 and 6. March 2023, respectively) and the scale factor for freezing ( $n_f$ ) over the hydrological year 2023.

2.1 Site description: It is recommended to specify the permafrost classification characteristics and thermal regime parameters in the study area. For instance: Permafrost type: continuous/discontinuous/sporadic/isolated patches. Mean annual ground temperature (MAGT) range, active layer thickness, permafrost thickness.

We agree with the referee and have included a description of permafrost characteristics and ground thermal conditions in response to a previous comment above.

**Line 134: What are the start and end months of the hydrological year?**

We agree with the referee that a definition of the hydrological year needs to be included. Indeed, the hydrological year differs substantially between regions and is typically chosen as to avoid the change of year to occur at times were substantial year-to-year variations in water storage is expected (e.g. snow or soil water). For similar reasons, we use a hydrological year where the change of year occurs on September 1st, which fits well with the times of our field visits and

ensures that the hydrological year contains a complete record of each 3-month season. In the revised manuscript we will include the following:

Line 134: Throughout this study we calculate temperature indices over the hydrological year starting on September 1st and ending on August 31st, meaning that the full hydrological years 2023 and 2024 are contained in the GST records.

Line 185: December represents the early snow accumulation phase, while March marks the end of winter when snow begins to melt. An additional snow measurement should be conducted in February during the stable snow cover period.

We agree with the referee that an additional snow measurement in February could have improved the presented dataset. Regrettably, a mid-winter field visit was no feasible within the constraints of the study. We did however not observe any signs of current or previous melt during the field visit in early March, indicating that our snow survey represents pre-melt conditions. Indeed, time series of snow depth from the nearby, forested Terelj site are presented in Dashtseren et al. (2014), which shows that snow cover from 2003-2007 builds up in response to a few snowfall events during winter, and that snow melt typically starts in March and lasts until April.

To clarify these points, we will add the following to the manuscript:

Line 75: Snow cover below the larch forest in Terelj typically last for 140-170 days with a maximum thickness of 8-18.6 cm in March (Dashtseren et al., 2014; Zweigel et al., 2024).

Figure 3: The authors have two years of data, so seasonal and annual averages should be presented with error bars. These error bars could help reveal certain patterns in the data. The authors have not provided complete figure titles for any of the figures, only including subfigure captions.

We agree that including error bars – showing spatial or year-to-year variability - could help analyse patterns in the presented data. However, our dataset consists of two years of 2-3 measurements per forest state, which is insufficient for spatially aggregated statistics. Indeed, our study focuses on the differences between forest states, which would be dwarfed by the annual variations if we e.g. combined the MAGST data for 2023 and 2024 in Fig. 3a. For this reason, we suggest keeping Figure 3 in its current form.

Regarding the figure captions, we have included a general title for all figures with subplots in response to a previous comment by the referee (see above).

**Line 272: Table C2?**

This is due to a typo, where we had included two "Table C1" in Appendix C. The second table will be relabelled **Table C2** in the revised manuscript.

**Lines 285-286: What's the reason? Such a result seems counterintuitive.**

We thank the referee for the attention to detail. During our field visits, we consistently find thinner surface organic layers in summer than in winter (Table C2), which are made up mostly by plant litter. We agree with the referee that this diminishing of surface organic layers is puzzling. Based on our observations of extensive livestock disturbances at sites with grass- and leaf litter in winter, we suggest livestock foraging as a potential explanation for the reduction in surface organic layer thickness. Furthermore, substantial decomposition of plant litter can occur

already within the first year, with higher decomposition rates for leaf- than needle-litter (Zhao et al., 2022). We also note that our observations of surface organic layers are done in late August/September, i.e. almost one year after their initial accumulation. Furthermore, we only measure the thickness of the organic layer during our field visits, not organic content, meaning that the thickness change could also partially be due to gradual settling of the surface organics.

In the revised manuscript, we will include the following clarification:

Line 368: This diminishing of surface organic layers throughout the year could be due to animal foraging of plan litter (see below), gradual settling, or decomposition which can be substantial already within the first year (Zhao et al., 2022).

Lines 291-296: This content has already been mentioned in the Introduction and Study Area sections, and its repetition here is redundant.

We agree with the referee and will remove it in the revised manuscript. We will move the first sentence to the second paragraph:

**4.1 Novel aspects and limitations**

This study presents a novel dataset of GSTs, vegetation density and surface conditions across a forest disturbance gradient at a site in the forest-steppe ecotone in central Mongolia. Based on these data, we investigate how forest disturbances impact vegetation and surface cover, and the direct and indirect affect that these disturbances have on GST dynamics. Previous studies from the forest-steppe ecotone of central Mongolia have shown a strong dampening of the annual GST cycle and a lowering of MAGSTs associated with forest cover (Dashtseren et al., 2014; Temuujin et al., 2024). However, the ecosystem gradients in these studies occur across substantial differences in topography, and the direct effect of forests on GSTs remains unclear. To untangle the impacts of terrain and ecosystems on ground thermal regime, our study specifically targets differences in forest states across sites with only small differences in topographic characteristics (Error! Reference source not found.). Overall, we find somewhat lower MAGSTs at disturbed sites than in intact forest, while the annual GSTs range is larger at sites where forest disturbances have reduced the canopy cover (Fig. 3).

Lines 367-368: What is the reason? Why has summer disappeared?

We agree with the referee that the reduction in litter layer thickness from winter to summer can be counterintuitive and hypothesise that this can be due to both consumption by livestock, settling of the litter throughout the seasons, and/or gradual decomposition throughout the growing season. Please see our reply and amendments to the manuscript in response to a previous comment on this matter.

Line 403: The interference may not necessarily lead to an increase in the MAGST, but it could still cause a significant rise in summer GST—it's just that the MAGST is offset by the decrease in winter GST.

Correct. In this paragraph we do discuss that despite forest disturbances in our study area giving higher summertime GSTs, these are outweighed by lower wintertime GSTs, resulting in somewhat lower MAGSTs at all disturbed forest states. In the revised manuscript we will more clearly present this interesting point by including the following amendment:

Line 399: Overall, our two years of measurements show an increased annual GST cycle and MAGSTs consistently 0.5°C lower at disturbed sites compared to intact forest (Fig. 3).

**Line 417: Why is that?**

Snow interception by vegetation is a process where taller and denser vegetation leads to preferential accumulation of snow in vegetated areas and on their lee sides. This is because vegetation interacts with the wind field, creating local areas of lower wind speed. During events of wind driven snow transport, snow erosion and deposition rates are driven by wind speed, with snow accumulating preferentially in areas with lower wind speeds. In areas with a variable vegetation cover and substantial wind speeds, this leads to a thicker snow cover in vegetated areas and in their wake. We note that no signs of wind redistribution were observed in the study area (see also next comment).

In the revised manuscript, we will add the following amendment for clarity:

Line 417: [...] snow capture, where interactions between wind and vegetation cover create areas of preferential snow accumulation at sites with higher and denser vegetation (Hiemstra et al., 2002; Sturm et al., 2001) [...]

Lines 415-429: Different vegetation cover can lead to changes in wind speed, and does wind speed affect snow cover? I believe livestock trampling might have some impact, but it's definitely not the main factor.

The referee is correct that differences in vegetation cover can induce local variations in wind speed, which can lead to small scale differences in snow erosion and deposition. Previous works (e.g. Hiemstra et al. 2002; Sturm et al. 2001) have found that at sites subject to substantial wind redistribution, snow preferentially accumulates under higher and denser vegetation canopies. Conversely, differences in vegetation cover can also lead to differences in interception loss, i.e. the amount of available snowfall that is intercepted by the canopy and subsequently sublimates back to the atmosphere, leading to lower snow depth below denser canopies (Fisher et al., 2016; Lundberg and Koivusalo, 2003; Yi et al., 2007). These interactions between vegetation cover and snow depth are discussed in lines 415-428 in the manuscript.

Regarding the presented study, our snow surveys in early and late winter show highly similar snow depths among the field sites, with no systematic differences between the forest states (Fig. C1). The correlation analysis also shows no strong correlation between snow cover and vegetation densities (Table D1). However, we agree with the referee that these surveys alone cannot eliminate the possibility of any wind-induced variations in snow depths in our study area. During our field visits, we did not observe any signs of wind-transported snow such as wind drifts, sastrugi or compacted surface snow layers. The only place where we observed a snow drift was immediately behind the ridge to the south-facing slope, which is outside our study area. We also note that the field sites are located in topographic depressions and concavities (Fig. 1) and thus are situated in areas that are relatively well sheltered against snow redistribution. In addition, the Siberian high-pressure system dominates the wintertime atmospheric conditions in central Mongolia, providing clear skies and weak winds (Ganbat and Baik, 2016; Hong et al., 2023). Combined, our observations and the general topographic and climatic conditions of the study area suggest that wind redistribution of snow is not a main factor controlling snow depths at our field sites.

In the revised manuscript, we will include the following passage in the discussion to clarify this point:

Line 419: Interestingly, we observe rather similar snow depths for the different forest states in Bayanzurkh (Fig. C1) and find no clear relationship between observed snow depth and  $n_f$  (Fig. C2) nor a strong correlation between snow depths and wintertime GSTs (Table D1). This lack of systematic differences in snow cover between the forest states could be linked to limited wind redistribution of snow due to overall sheltered location of the field sites (Sect. 2.1) and the generally low wintertime windspeeds in central Mongolia (Hong et al., 2023), and the deciduous nature of the forest in our field area, which limits snow interception.

Line 433-434, Table C2: How do you determine whether livestock trampling is sparse or dense? By the number of hoofprints? But could there be cases where heavy trampling is later covered by snow?

We agree with the referee that this matter should be clarified. We classify the degree of disturbance by livestock as "scattered" or "extensive" based on the fraction of the snow surface that is visually impacted by livestock at the time of our field visits. The disturbances include areas that have been trampled, where livestock have laid down/rested, and where the snowpack has been reworked (likely by livestock in search of feed). In the caption of Table C2, we describe the classifications: "Trampling is characterized as "scattered" and "extensive" if respectively less than 10% or more than 50% of the snow surface are visually disturbed by livestock". We do however acknowledge that such a classification is subject to observer subjectivity but argue that there is a noticeable distinction between these two degrees of disturbance. In Figure 1a and b we showcase typical snow conditions in the region that we would classify as "scattered" and "extensive" livestock disturbance. Regrettably, we were not able to obtain any usable images of snow conditions at the Bayanzurkh study area during our field visits in winter.

Figure 7: Examples of typical snow conditions at a nearby sites in central Mongolia in March 2023. a) and b) show snow cover where livestock disturbance would be characterised as "scattered" and "extensive", respectively. c) shows a snow profile that consists of depth hoar and rounded facets throughout, without any clear snow layering.

Snow infilling of previously trampled areas is certainly possible. Indeed, we did observe abrupt changes in snow hardness at some locations during our snow survey in March 2023, which could be due to light fresh snow overlying previously trampled snow. However, quantifying the spatiotemporal patterns of livestock trampling and snow cover evolution would require a nearcontinuous presence at the study site throughout the cold season, which was not possible within the scope of the study. Consequentially, we are limited to a qualitative description of the livestock disturbances at the times of our field visits. Here, we want to point out that we were struck by the systematic differences in disturbance degrees at our field visits. In late December, we only observed a few animal tracks traversing the field sites, while in early March we encountered a snow cover that was nearly completely reworked by livestock activity at many of the field sites (see examples in Figure 1a and b above). At several sites the degree of disturbance was so large that we were not able to find any suitable locations for a measurement of "undisturbed" snow cover depth. For this reason, we took several snow depth measurements at each field site sampling the apparent variability in snow cover and present the average depth of these in the manuscript (see description in Sect. 2.4 and captions of Fig. C1 and Table C2). Notably, we observed "extensive" trampling at all sites with disturbed forest and in the aspen stand, while we found only "scattered" trampling at the intact larch forest and in the stand of young larch (Table C2). This systematic difference in livestock activity could be due to the availability of edible grass and leaf litter at the disturbed sites, or that livestock and herders prefer winter feeding grounds with a more open canopy.

We acknowledge that our classification of livestock disturbance to the snow cover as "scattered" and "extensive" is not suitable for quantitative assessment the role of livestock activity in shaping wintertime GST dynamics (e.g. inclusion in the correlation analysis). We do nevertheless think that it is necessary to include these observations in the manuscript, as removal or reduction of the snow cover can have strong local impact on the investigated GSTs. While we do not have direct measurements relating disturbance of the snow cover and

wintertime GSTs, we find sudden divergences in GST for some field sites in winter. These GST divergence events occur at various times at different sites and are typically followed by colder GSTs throughout the remainder of the winter period. We also note that such divergences in GST are found in the temperature records of one or several of the field sites in the dead and logged forest in both 2023 and 2024 (as well as the young aspen stand), while they are absent in the intact forest and the stand of young larch. We agree that direct linking of the observed livestock disturbance and GST divergence is not possible with the available data. We do however argue, given the widespread nature and systematic differences in livestock disturbances, that this factor needs to be at least qualitatively discussed in the manuscript.

In the revised manuscript we will make the following amendments to improve clarity and highlight these matters:

Line 187: We also qualitatively assess the extent of livestock disturbance to the snow cover at the time of visit, which we classify as "scattered" and "extensive" if less than 10% or more than 50%, respectively, of the snow surface is visibly disturbed by livestock activity.

Moving table C2 to the main text (new Table 3):

Table 1: Observations of snow cover at the individual field sites. Note that the snow depths on 06.03.2023 are the mean values of several snow measurements. Livestock disturbance is characterized as "scattered" and "extensive" if respectively less than 10% or more than 50% of the snow surface are visually disturbed by livestock activity.

|          | · · · · · ·  |            |             |            |             |
|----------|--------------|------------|-------------|------------|-------------|
| Logger   | Forest state | 22.12.2022 |             | 06.03.2023 |             |
|          |              | Snow depth | Disturbance | Snow depth | Disturbance |
| INTACT_1 | Intact       | 21 cm      | Scattered   | 18 cm      | Scattered   |
| DEAD_1   | Dead         | 10 cm      | Scattered   | 19 cm      | Extensive   |
| DEAD_2   | Dead         | 19 cm      | Scattered   | 16 cm      | Extensive   |
| YOUNG_1  | Young        | 21 cm      | Scattered   | 20 cm      | Extensive   |
| YOUNG_2  | Young        | 18 cm      | Scattered   | 17 cm      | Scattered   |
| DEAD_3   | Dead         | 16.5 cm    | Scattered   | 20 cm      | Extensive   |
| LOGGED_1 | Logged       | 20 cm      | Scattered   | 15 cm      | Extensive   |
| LOGGED_2 | Logged       | 21.5 cm    | Scattered   | 16 cm      | Extensive   |
| LOGGED_3 | Logged       | 17 cm      | Scattered   | 14 cm      | Extensive   |
| INTACT_2 | Intact       | 20 cm      | Scattered   | 16 cm      | Scattered   |

New discussion paragraph in Section 4.1:

More systematic surveys of snow cover are required for quantifying its impact on GST dynamics. The survey in late winter reveals a snow cover that is substantially disturbed by livestock at several of the field sites, especially in the dead and logged forest (Error! Reference source not found.). We classify this disturbance as "scattered" and "extensive" based on the percentage of snow surface that is visually disturbed by livestock activity (Sect. 2.4). Such a classification is however not suitable for the correlation analysis (Sect. 3.4), and our further discussion of the role of livestock disturbances in shaping wintertime GSTs remains qualitative. Furthermore, our snow sampling does not capture events where previously disturbed areas are infilled by subsequent snowfalls. However, a more comprehensive and quantitative assessment of livestock disturbance was not possible within the scope of this study. Future research efforts could for example use repeated airborne LiDAR surveys in winter for detailed snow depth mapping (Koutantou et al., 2022),

which could provide information on the snow depth evolution at the location of the GST measurements, and be used to analyse the extensiveness of snow trampling within each forest state.

New Figure B1 showing an example of a GST divergence event and reference to this figure in the text:

Line 205: [...] and in winter and spring where we observe sudden divergence in GSTs at sites subject to livestock trampling (Sect. 3.4, Fig. B1).

Figure B1: Evolution of daily GSTs for the individual loggers in the young forest stands winter 2024. The arrow indicates a time in late December from which the two loggers swich from having highly similar GSTs to having strongly diverging GSTs until late winter, possibly in response to livestock trampling at the young aspen site (YOUNG\_1).

Line 290: The authors should construct a relational diagram between all factors and GST (Ground Surface Temperature), calculate the contribution rate of each factor to GST, and then discuss which one is the dominant influencing factor. Otherwise, the discussion can only superficially address whether the factors affect GST and which one plays a major role. Due to the presence of anthropogenic factors such as livestock trampling, it is impossible to analyze GST differences caused solely by variations in vegetation cover.

We thank the referee for this suggestion. In response to a previous comment, we have performed a correlation analysis between GSTs and variables relating to vegetation density, snow cover, surface organic layers and terrain characteristics. This analysis revealed that vegetation density (specifically PAIcanopy) is strongly correlated to seasonal GSTs, whereas any strong correlations with the other environmental variables are absent. Here, we point out that the sites are chosen to have similar terrain characteristics. As this analysis so clearly shows that vegetation density is the dominant factor influencing GSTs at our field sites, we feel that constructing a relational diagram would be superfluous. We also agree with the referee that the influence of livestock trampling on GSTs cannot be quantitatively assessed with the available dataset and have added a discussion of this limitation in response to a previous comment.

Line 464: I believe such a conclusion is unreasonable. Livestock trampling compacts snow cover, reducing its thickness while increasing its density. However, this factor has not been

quantified, nor has the influence of snow layers—another critical aspect—been quantitatively assessed.

We agree with the reviewer that such a conclusive statement about the role of livestock should be removed from this paragraph.

Regarding snow layers, no distinct snow layers were found during the snow surveys. Indeed, the snow cover consisted almost entirely of chains of depth hoar and strongly faceted crystals (Fig. 1c above). The snow grain size was measured at some sites and found to be typically 5-7 mm at the base of the snowpack and 2-4 mm towards the surface. The presence of depth hoar/facets throughout the snowpack is indicative of strong vapor pressure gradients which drives constructive snow metamorphism. The persistent high-pressure systems that shape winter climate in Mongolia provide generally low snowfall amounts and cold air temperatures, creating a strong temperature gradient through the thin snowpack. Along this gradient, water vapor is transported from warmer areas near the ground surface towards colder areas above, driving sublimation at the top of snow grains and deposition on their base. Over time, this process leads to a growth of well-oriented hoar grains and the diminishing of any layering created during snow deposition. Furthermore, the persistent low temperatures and weak winds that characterize Mongolian winters inhibit the establishment of melt layers or wind compacted layers within the snowpack.

To address these matters raised by the referee, we will include the following amendments to the manuscript:

Line 271: The snow cover at all field sites consisted of depth hoar and faceted crystals with no clear horizontal layering.

Line 463: Overall, we find about 0.5°C lower surface offset (Eq. 1) in the dead and logged forest and stands of young regrowth compared to the intact forest (Table 2), indicating a general cooling of the ground surface following forest disturbance.

Line 475: The understory vegetation in logged or dead forest plots is dense, resulting in a Plant Area Index (PAI) similar to that of intact forests. However, their effects on ground surface temperature (GST) still differ. Relying solely on PAI may fail to distinguish the varying impacts of different vegetation covers on GST.

We agree with the referee that floor vegetation and forest canopies have different impact on GSTs. For exactly this reason, our analysis distinguishes between PAItotal, which is derived from hemispheric images taken at the ground surface and includes contribution from both floor vegetation and the forest canopy, and PAIcanopy which only includes the forest canopy. The difference in GSTs between sites with similar PAItotal but different PAIcanopy is discussed in Section 4.3 (Lines 378-88). In the current paragraph, we point out that these differences in structure are not necessary captured by spectral remote sensing products, which limits their applicability in detecting forest disturbances and their impact on GST dynamics.

To improve and clarify this, we will make the following amendments in the manuscript:

Line 376: This suggests that vegetation indices derived from spectral remote sensing products might be inadequate for detecting or mapping the spatial extent of such forest disturbances and for assessing their impact on GSTs.

[revised manuscript text omitted]